# Patient-derived xenografts and single-cell sequencing identifies three subtypes of tumor-reactive lymphocytes in uveal melanoma metastases

Joakim W Karlsson[1,2†], Vasu R Sah[2†], Roger Olofsson Bagge[2,3,4], Irina Kuznetsova[1], Munir Iqba[5], Samuel Alsen[2], Sofia Stenqvist[2], Alka Saxena[5], Lars Ny[2,6], Lisa M Nilsson[1,2], Jonas A Nilsson[1,2*]

[1]Harry Perkins Institute of Medical Research and University of Western Australia, Perth, Australia; [2]Sahlgrenska Center for Cancer Research, Institute of Clinical Sciences, Sahlgrenska Academy, University of Gothenburg, Gothenburg, Sweden; [3]Department of Surgery, Sahlgrenska University Hospital, Gothenburg, Sweden; [4]Wallenberg Centre for Molecular and Translational Medicine, University of Gothenburg, Gothenburg, Sweden; [5]Genomics WA, Telethon Kids Institute, Harry Perkins Institute of Medical Research and University of Western Australia, Nedlands, Australia; [6]Department of Oncology, Sahlgrenska University Hospital, Gothenburg, Sweden

*For correspondence:
jonas.a.nilsson@surgery.gu.se

†These authors contributed equally to this work

**Abstract** Uveal melanoma (UM) is a rare melanoma originating in the eye's uvea, with 50% of patients experiencing metastasis predominantly in the liver. In contrast to cutaneous melanoma, there is only a limited effectiveness of combined immune checkpoint therapies, and half of patients with uveal melanoma metastases succumb to disease within 2 years. This study aimed to provide a path toward enhancing immunotherapy efficacy by identifying and functionally validating tumor-reactive T cells in liver metastases of patients with UM. We employed single-cell RNA-seq of biopsies and tumor-infiltrating lymphocytes (TILs) to identify potential tumor-reactive T cells. Patient-derived xenograft (PDX) models of UM metastases were created from patients, and tumor sphere cultures were generated from these models for co-culture with autologous or MART1-specific HLA-matched allogenic TILs. Activated T cells were subjected to TCR-seq, and the TCRs were matched to those found in single-cell sequencing data from biopsies, expanded TILs, and in livers or spleens of PDX models injected with TILs. Our findings revealed that tumor-reactive T cells resided not only among activated and exhausted subsets of T cells, but also in a subset of cytotoxic effector cells. In conclusion, combining single-cell sequencing and functional analysis provides valuable insights into which T cells in UM may be useful for cell therapy amplification and marker selection.

## eLife assessment

This study presents **valuable** findings on tumor-reactive T cells in liver metastases of uveal melanoma (UM). The authors conducted single-cell RNA sequencing to identify potential tumor-reactive T cells and used PDX models for functional analysis. The evidence supporting their claims is **solid**. The work will be of interest to scientists working in the field of uveal melanoma.

## Introduction

Uveal melanoma (UM) is a rare form of melanoma and the most common primary malignancy of the eye (*Jager et al., 2020*). It develops in the uvea of the eye, most often in the choroid and ciliary body, and more infrequently in the iris. Primary UM can be cured by brachytherapy or enucleation of the eye, but 50% of patients will develop metastasis (*Buder et al., 2013*). The most common route for meta-static spread is the liver for reasons that are poorly understood. Patients with metastatic disease have a median survival of less than a year (*Khoja et al., 2019*) however, ongoing clinical studies suggest that some treatments could prolong survival (*Carvajal et al., 2017*). Monotherapy immune checkpoint inhibitors (ICI) are markedly less effective in patients with UM (*Johnson et al., 2019*; *Algazi et al., 2016*) than in those with metastatic cutaneous melanomas. However, combination treatments with PD-1/CTLA-4 inhibitors (*Piulats et al., 2021*; *Pelster et al., 2021*; *Najjar et al., 2020*) or PD-1/HDAC inhibitors (*Jespersen et al., 2019*; *Ny et al., 2021*) have demonstrated longer overall survival than historic benchmark data (*Khoja et al., 2019*). Tebentafusp, a T cell engager suitable for patients with the HLA-A2 genotype (*Damato et al., 2019*; *Middleton et al., 2020*), is the first therapy to show a prolonged overall survival in a phase 3 randomized trial, increasing median survival from 16.0 months to 21.7 months (*Nathan et al., 2021*). Another development is locoregional treatments, where a recent phase 3 trial demonstrated that isolated hepatic perfusion (IHP) triples hepatic progression-free survival (*Olofsson Bagge et al., 2023*) compared with the best alternative care and historic benchmark data. Although retrospective data suggest an overall survival benefit with liver-directed therapy (*Ben-Shabat et al., 2016*), in the SCANDIUM phase 3 trial, overall survival of patients treated with IHP was not statistically significantly better that for patients treated with best alternative care (*Olofsson Bagge et al., 2024*). Moreover, invariably, all patients progressed on both tebentafusp and IHP; therefore, more research is needed.

Adoptive cell therapy (ACT) with TILs or CAR-T cells has not been extensively studied in UM. Pilot data from a clinical trial (*Chandran et al., 2017*) showed that TILs can cause responses in patients, and we have previously showed that HER2 CAR-T cells can eradicate UM in xenografts (*Forsberg et al., 2019*). There is, however, a lack of robust ex vivo screening models and very few patient-derived xenograft (PDX) mouse models from metastases (*Carita et al., 2015*) to use in studies to improve ACT. Part of the lack of success for ACT may be attributable to a lack of tumor-reactive lymphocytes, possibly owing to lower mutation burden in UM compared to cutaneous melanoma (*van der Kooij et al., 2019*). However, both the clinical trial and previous analyses of TILs in UM liver metastases have suggested that at least some TILs are tumor reactive (*Chandran et al., 2017*; *Durante et al., 2020*; *Karlsson et al., 2020*; *Rothermel et al., 2016*).

Defining tumor-reactive T lymphocytes (TRLs) among TILs would enable the identification of candidate biomarkers for selective expansion or TCR transgenics in cell therapy experiments in mouse models and cell therapy trials. To decipher how immunotherapy can be made more effective, the aim of this study was to identify and functionally validate TRLs in metastases of patients with UM. To this end, we used paired single-cell RNA (scRNA)- and TCR-seq, as well as functional experiments using PDX models and autologous TIL cultures.

## Results

We previously collected metastases of UM from patients in routine clinical care and two clinical trials. These were the SCANDIUM phase 3 randomized trial comparing IHP to best alternative care (*Olofsson et al., 2014*) and the PEMDAC phase 2 single-arm trial where patients received a combination of the PD-1 inhibitor pembrolizumab and the HDAC inhibitor entinostat (*Ny et al., 2021*). Owing to the limited amount available of these biopsies, we prioritized one part of the biopsy for DNA/RNA preparation to sequence (*Karlsson et al., 2020*), one part for transplantation into immunocompromised mice to generate PDX mouse models, one part for generation of TILs, one part for cryopreservation of finely minced tumor (flow cytometry and single-cell sequencing), and one part for formalin-fixed paraffin-embedded (FFPE) blocks (*Figure 1a*).

### Single-cell sequencing defines an atlas of T cell states in UM metastases

We obtained biopsies suitable for sequencing from 14 patients, two of which were subcutaneous and 12 were liver metastases. To identify and profile the T cells present in these samples, we performed

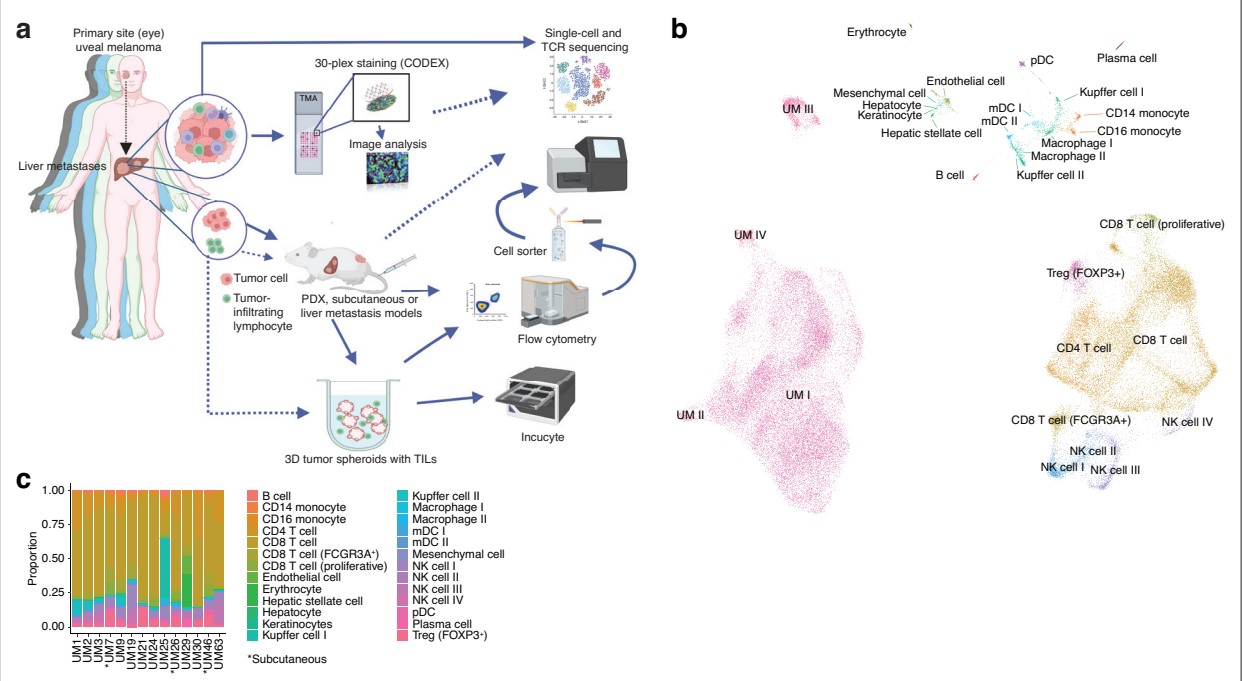

**Figure 1.** Single-cell RNA sequencing of uveal melanoma metastases. (**a**) Schematic showing the study workflow. Tumors extracted from patient liver metastases were subjected to single-cell RNA-seq (scRNA-seq), formalin fixation for immunohistochemical (IHC) analysis, and patient-derived xenograft (PDX) generation. (**b**) Uniform Manifold Approximation and Projection (UMAP) dimensionality-reduced expression profiles of single cells in biopsies from 14 patients with uveal melanoma (UM). scRNA-seq data from all samples were integrated using FastMNN, Louvain clustering performed, and clusters annotated using cell-type marker genes compiled from the literature. Some cell types were annotated after additional subclustering (**Figure 1—figure supplement 1a–f**). (**c**) Cluster proportions within each sample, for all cell types. Subcutaneous biopsies are indicated with asterisk, remaining samples being liver biopsies. Panel a generated with BioRender.com.

The online version of this article includes the following figure supplement(s) for figure 1:

**Figure supplement 1.** Identification of cell types in single-cell RNA sequencing data.

paired scRNA- and TCR-seq, integrated the samples, and annotated the cell types. This has resulted in the largest single-cell atlas of metastatic UM to date, complementing prior efforts, including our own study of TILs from UM metastases (**Durante et al., 2020**; **Karlsson et al., 2020**; **Lin et al., 2021**; **Pandiani et al., 2021**).

Uniform Manifold Approximation and Projection (UMAP) dimensionality-reduced expression profiles revealed clusters containing melanoma, CD8$^+$ or CD4$^+$ T cells, NK cells, monocytes, and small clusters of B cells, endothelial cells, and other minor cell populations (**Figure 1b and c**, **Figure 1—figure supplement 1a–f**). Clusters were generally well mixed with respect to contributions from different samples, suggesting few patient-specific or batch effects after data integration (**Figure 1—figure supplement 1g**). The identity of cells labeled as UM was further confirmed by inferring copy number changes. This revealed characteristic aberrations seen in metastatic UM, including monosomy of chromosome 3 and gain of 8q, as well as subclonal variation within patients (**Figure 1—figure supplement 1h**).

Further subclustering of CD8$^+$ T cells showed the presence of 12 different groups (**Figure 2a and b** and **Figure 1—figure supplement 1i–k**), all characterized by the differential expression of immunological marker genes (**Figure 2c and d** and **Supplementary file 1**). The clusters can be divided into two axes. One of these is characterized by expression of genes associated with late activation and exhaustion, such as *PDCD1*, *HAVCR2*, *TNFRSF9*, and *HLA-DRA*. Additional expression of *ITGAE* (CD103) suggests that these clusters contain tissue-resident memory cells (**Ganesan et al., 2017**). Multiplex immunofluorescence showed that PD-1- and ICOS-expressing T cells were in close proximity to the tumor cells (**Figure 2e**). The other axis expressed genes associated with naïve/memory-like or early activated phenotypes, such as *IL7R*, *TCF7*, *CCR7*, and *NR4A1* (**Figure 2d**).

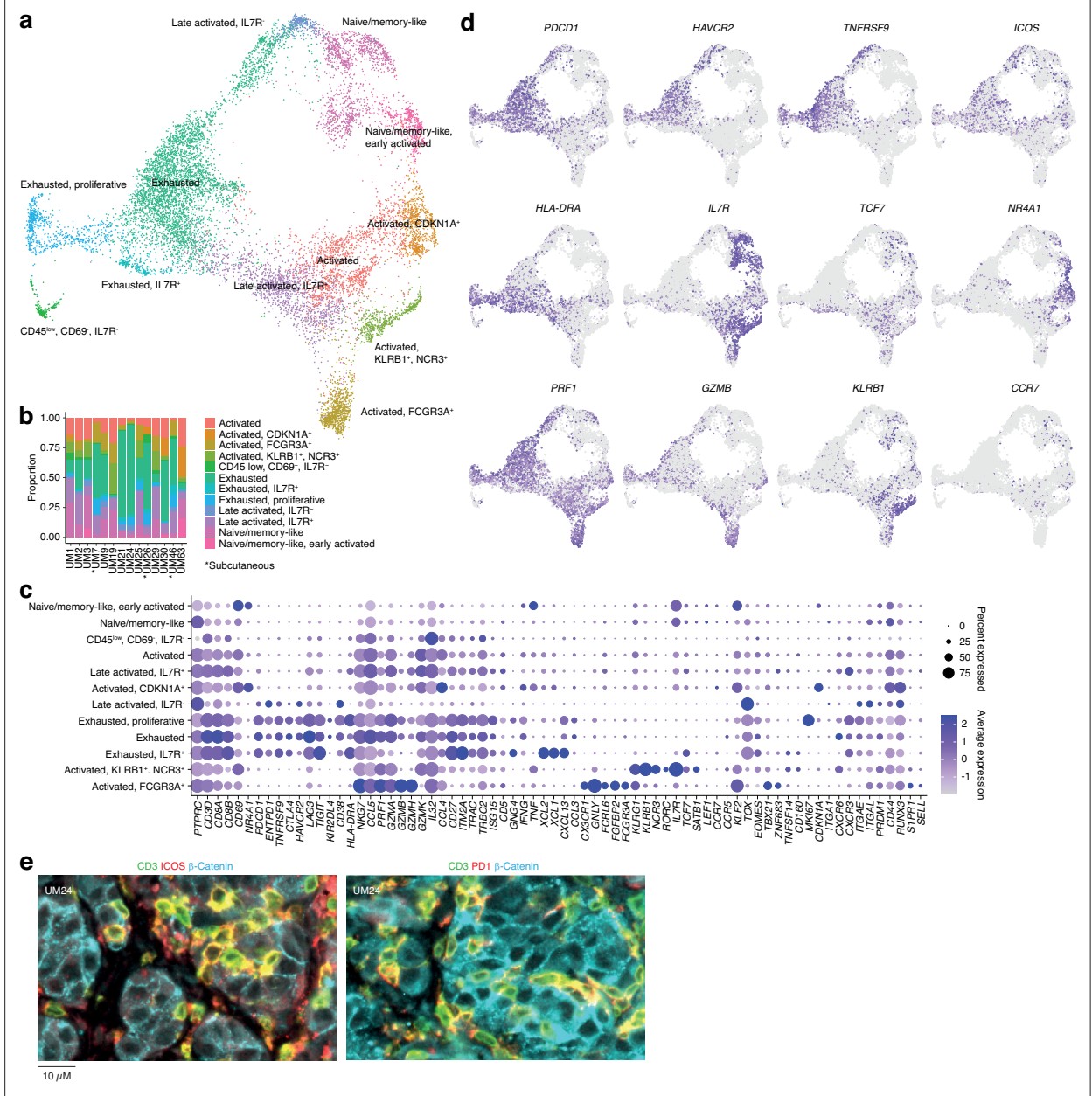

**Figure 2.** CD8 T cells in uveal melanoma metastases. (**a**) Uniform Manifold Approximation and Projection (UMAP) dimensionality reduction and more detailed re-annotation of the CD8[+] T cell subsets from *Figure 1b*, using the marker genes shown in (**c**). (**b**) CD8[+] T cell cluster proportions within each sample. (**c**) Expression levels of markers used for CD8[+] T cell subset annotation. (**d**) Expression of marker genes signifying early (*NR4A1*) activation, naïve/memory-like phenotypes (*TCF7, IL7R*), late activation (*HLA-DRA, PDCD1, ICOS*), and progression toward exhaustion/dysfunction (*LAG3, TIGIT, HAVCR2*). Additional statistically identified genes differing between clusters can be found in *Supplementary file 1*. (**e**) Multiplex immunohistochemical (IHC) staining for T cell (CD3, ICOS, PD-1) and cancer marker genes (β-catenin) in a patient biopsy (UM24).

Individual clusters within these axes mostly correspond to intermediate phenotypes that progress from activation to exhaustion. However, a few seem to represent distinct states. Two of these were within the *IL7[+]* axis. Of these, one expressed *KLRB1* and *NCR3* and the other expressed NK-associated markers, such as *FCGR3A, FCRL6,* and *NKG7* (*Figure 2c and d*). Although both clusters expressed genes related to cytotoxicity, only the latter expressed *GZMB* and *GZMH*. Other markers associated with this group are *CX3CR1* and *FGFBP2*. Previous studies labeled these cells as cytotoxic NK-like T cells (*Li et al., 2019*; *Huuhtanen, 2023*).

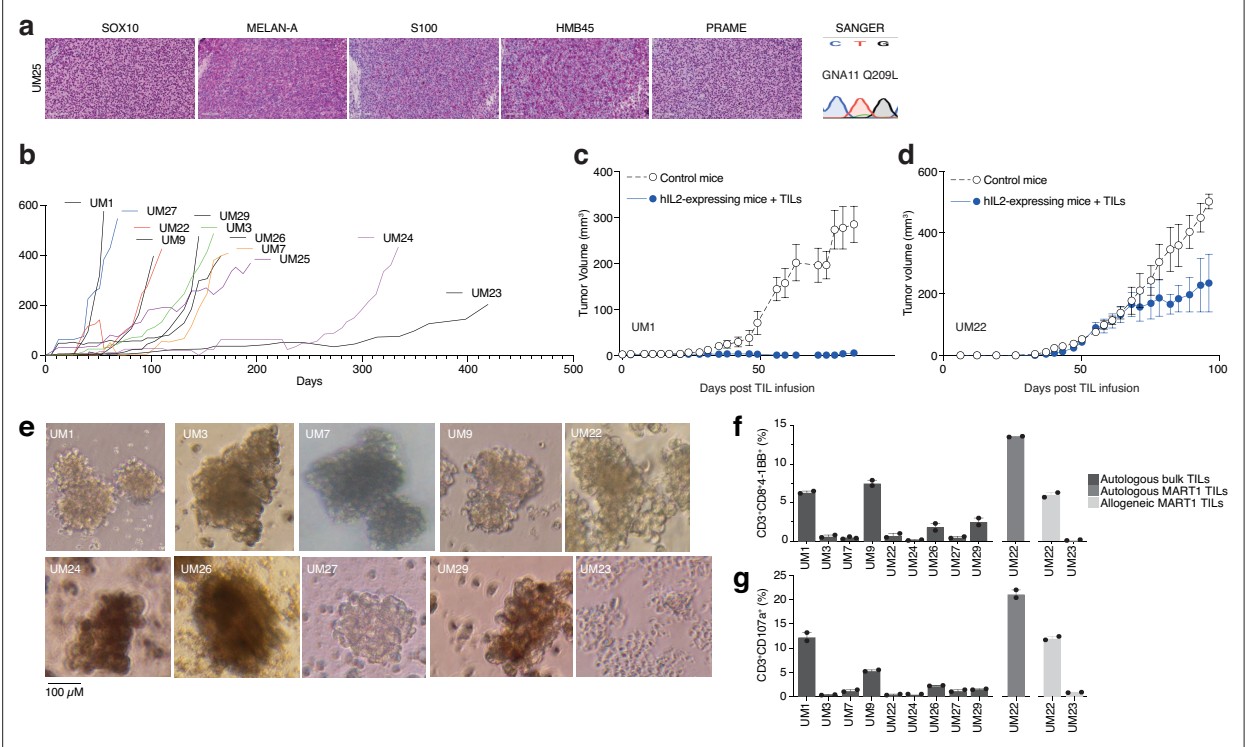

**Figure 3.** PDX models and 3D tumorsphere cultures of uveal melanoma to find TRLs. (**a**) Immunohistochemical (IHC) staining of HMB45, SOX10, S100, MELAN-A (MART1), and PRAME expression in one patient-derived xenograft (PDX) (see also *Figure 1—figure supplement 1* for additional PDX models), as well as Sanger sequencing to verify mutation of *GNA11* at Q209L. Color reactions were with magenta. (**b**) Tumor growth curves of established PDX models. (**c–d**) Tumor-infiltrating lymphocytes (TIL) therapy experiments in PDX models. UM1 and UM22 were transplanted subcutaneously into NOG or hIL2-NOG mice. When tumors were palpable, 20 million autologous TILs were injected in the tail vein. Tumor growth was monitored by a caliper. Tumor sizes are average size ± SEM. (**e**) Images of tumor spheroids created (scale bar represents 100 µM). UM23 was unable to form spheres. (**f–g**) Activation (41BB⁺) and degranulation (CD107a⁺) status for spheres and TIL co-cultures, (**f**) and (**g**) respectively, using n=2 or 3 replicates.

The online version of this article includes the following figure supplement(s) for figure 3:

**Figure supplement 1.** Immunohistochemical (IHC) showing expression of HMB45, SOX10, S100, MELAN-A, PRAME expression in patient-derived xenograft (PDX) models.

**Figure supplement 2.** Characterization of TILs used in sphere culture experiments.

---

In summary, this map reveals the cell types present in liver and subcutaneous metastases of UM and divides infiltrating T cells into two major phenotypic groups, characterized by the mutually exclusive expression of *HAVCR2* and *IL7R*, but also reveals distinct cytotoxic subsets expressing NK-associated markers.

## TILs from UM metastases can kill their cognate tumors in PDX models

Given the presence of activated T cells in tumors, we investigated whether these could eradicate metastatic UM in ACT experiments using PDX models. These models are useful personalized models for studying tumor biology and more recently, tumor immunology (*Patton et al., 2021*). However, the biopsies we received were generally very small, and the initial take rate using the same single-cell dispersion method used for cutaneous melanoma metastases (*Patton et al., 2021*) was unsatisfactory. Therefore, we transplanted small tumor pieces or used splenic injections, which led to successful PDX generation for some patients. Ten of these are presented in this study. All the PDX tumors exhibited confirmatory markers of melanoma, such as SOX10, MELAN-A (MART1), HMB45, or S100B. Sanger sequencing further revealed mutually exclusive mutations in the UM oncogenes *GNAQ*, *GNA11*, *CYSLTR2,* and *PLCB4*, as found in biopsies (*Karlsson et al., 2020*; *Figure 3a* and *Figure 3—figure supplement 1*). The UM tumors grew at a slower rate compared to our experience with cutaneous

melanoma, which has a take rate of 95% and where a model is usually established within 3–12 weeks (*Figure 3b*; *Einarsdottir et al., 2014*).

We have previously developed a method to study ACT in PDX models, termed PDXv2 (*Forsberg et al., 2019*; *Jespersen et al., 2017*; *Webb et al., 2010*). Survival, expansion, and activity of TILs and CAR-T cells were improved in this model using human IL2-expressing transgenic NOD SCID IL2 receptor gamma knockout mice (hIL2-NOG) (*Forsberg et al., 2019*; *Jespersen et al., 2017*; *Ny et al., 2020*; *Huuhtanen et al., 2022*; *Nilsson et al., 2022*). Tumor eradication by TILs correlates with immunotherapy response in patients with melanoma (*Jespersen et al., 2017*; *Ny et al., 2020*). We conducted a PDXv2 experiment using PDX models from two patients with UM. The injection of autologous TILs into hIL2-NOG mice carrying UM1 resulted in tumor rejection, whereas UM22 tumors could not be eradicated (*Figure 3c and d*). This constitutes proof of concept that TILs from UM metastases can kill autologous tumors in a PDXv2 model.

## TILs contain subsets capable of ex vivo expansion and killing in 3D tumor sphere cultures

To gain further insight into the capacity of T cells in UM biopsies to kill tumors, we next investigated the tumor reactivity of ex vivo expanded TILs on autologous UM cells. Since most UM PDX models did not grow fast enough to routinely conduct ACT experiments, we used the tumors grown in the mice to generate 3D tumor sphere cultures (*Ma et al., 2021*; *Weiswald et al., 2015*) of high viability for TIL experiments (*Gopal et al., 2021*; *Al Hity et al., 2021*). We generated tumor sphere cultures from 10 PDX models and expanded their autologous TILs (*Figure 3e*). When tumor spheres and TILs were co-cultured, TILs from patients UM1 and UM9 were activated the most (surface 4-1BB$^+$) and also degranulated (surface CD107a$^+$; *Figure 3f and g*). These TILs also bound to the tumor spheres, as assessed by labeling tumor spheres and TILs with different vital dyes and then imaging with a real-time cell culture imager (*Figure 3—figure supplement 2*). The UM22 TILs were hardly able to get activated by their autologous tumor spheres (*Figure 3b*), which likely explains their lack of activity in the PDX ACT experiment (*Figure 3d*). However, since UM22 is capable of presenting MART1 on HLA-A2, and since we previously identified a minor population of MART1-reactive TILs in this biopsy (*Karlsson et al., 2020*), we decided to expand and enrich this subset using MART1 peptide HLA-A2 dextramers. When using these cells, we were able to achieve potent activation and degranulation (*Figure 3f and g*). This confirms that T cells extracted from biopsies can contain subsets capable of recognizing and killing tumors, which can be expanded ex vivo.

## Determining the phenotypes of TRLs in biopsies using scRNA- and TCR-seq

To gain clues on how to identify reactive T cells from biopsies through their transcriptomic phenotypes, we co-cultured UM1 and UM9 TILs with their autologous tumors and sorted 4-1BB activated T cells by cell sorting. We prepared RNA from these cells, and as a control, from MART1-specific allogenic UM46 TILs and sequenced their TCRs. Then, using the single-cell atlas built from biopsies, we mapped those TCRs back to T cell subsets from their original tumors. This showed that those TRLs predominantly emanated from exhausted and late-activated cells (*Figure 4a and b*). UM9 also contained a sizeable amount of TRLs in the NK-like/cytolytic cluster (*Figure 4b*). The lack of potential TRL clones identified in the *TCF7*$^+$ axis of the CD8$^+$ T cell atlas from biopsies is also compatible with a previous report showing that *TCF7*$^+$ T cells in melanoma tend to be non-reactive bystanders (*Li et al., 2019*).

To evaluate which T cells in biopsies were reflective of those capable of generating TIL cultures, we further mapped our previously published TCR-seq data of UM TILs to the biopsy data (*Figure 4c*) to compare them to the clones matching experimentally identified TRLs in the biopsy (*Figure 4b–d*, *Figure 4—figure supplement 1a and b*, and *Supplementary file 2*). Cells from the TIL dataset tended to correspond to biopsy T cells with exhausted and late-activated profiles, similar to the experimentally identified TRLs. This might be expected since both have undergone expansion, which might favor T cells in certain phenotypic states. However, TRLs were significantly over- and underrepresented in some phenotypic clusters compared to TILs in general. Overrepresented clusters included proliferative exhausted and *IL7R*$^+$ exhausted T cells. The underrepresented ones were activated (non-exhausted), naïve/memory-like, and *FCGR3A*$^+$ NK-like T cells. The distribution of TRLs across the clusters was also

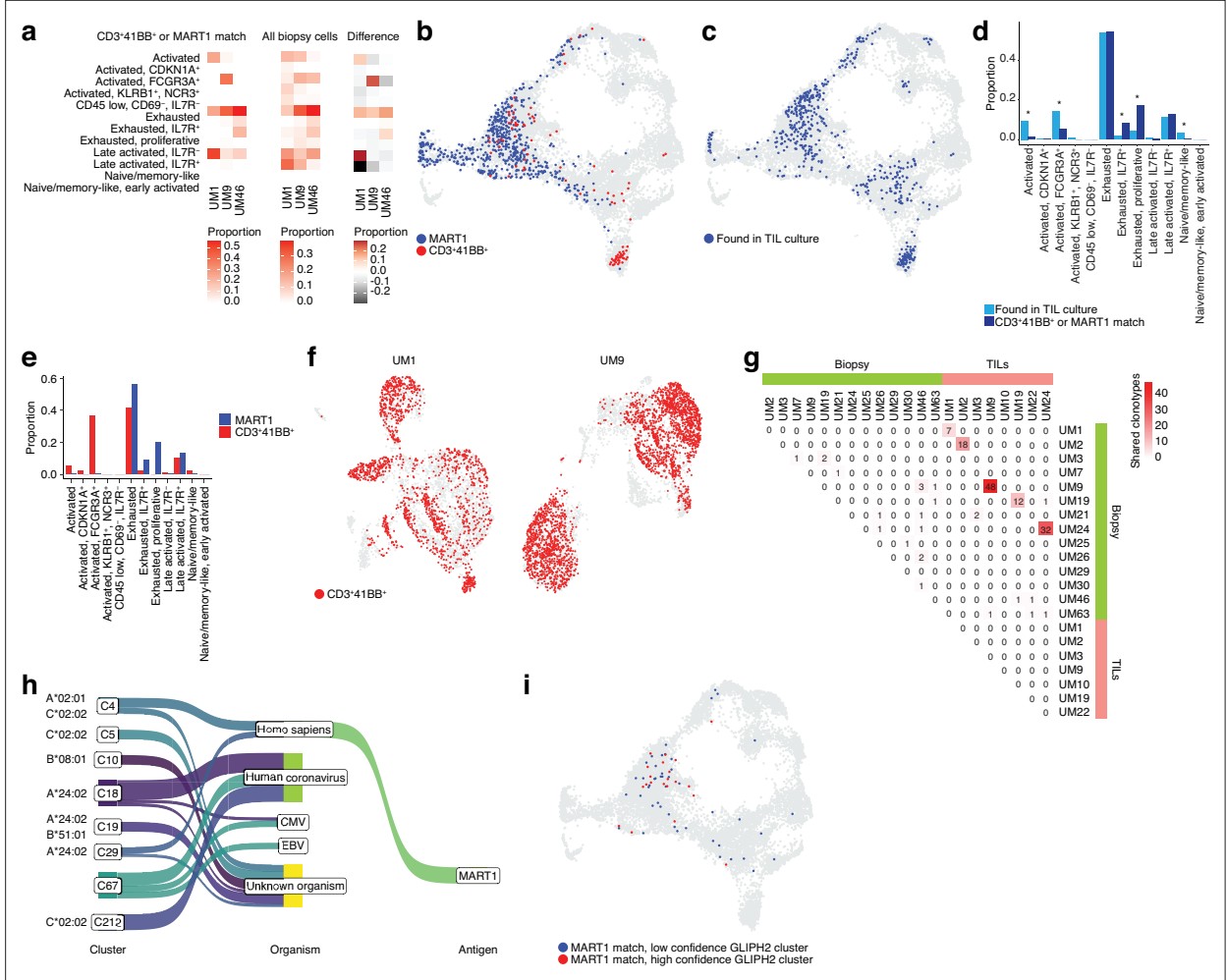

**Figure 4.** TCR sequencing of CD8 T cells. (**a**) Proportions of cells from each sample that match clonotypes found in experimentally identified 4-1BB⁺ and MART1-reactive T cells and which are present in a given biopsy CD8⁺ T cell cluster. The difference between matching and non-matching cells in each cluster is shown on the right, highlighting subsets that are enriched among the reactive T cells. (**b**) Biopsy CD8⁺ T cells with clonotypes matching identified reactive T cells, highlighted in the Uniform Manifold Approximation and Projection (UMAP) representation. (**c**) As in (**b**), but highlighting cells with clonotypes shared between biopsies and tumor-infiltrating lymphocytes (TILs). (**d**) Proportions of biopsy CD8⁺ T cells in each cluster matching either clonotypes from TIL cultures or experimentally identified reactive T cells. Statistical differences in frequency were determined using binomial tests between the frequency of the latter in each cluster relative to the background frequency of all TIL clonotypes present in the same cluster. Frequencies were calculated as number of cells from a given category in a given cluster divided by the total number of cells from that category, where category refers to either TILs or 4-1BB⁺/MART1-reactive cells. p-Values were adjusted for multiple testing using Bonferroni correction. (**e**) Distributions of cells matching the two different categories of experimentally identified reactive cells among biopsy CD8⁺ T cells clusters. (**f**) Cells from single-cell RNA-seq (scRNA-seq) of TIL cultures that have clonotypes matching experimentally identified reactive cells. (**g**) Shared clonotypes among all biopsy and TIL samples. (**h**) GLIPH2 clusters of significantly similar and HLA-restricted clonotypes (*Huang et al., 2020*), mapped to known antigens in public databases using TCRMatch (*Chronister et al., 2021*), based on CDR3β sequences. (**i**) Biopsy CD8⁺ T cells with clonotypes in either high- or low-confidence GLIPH2 clusters that match MART1 motifs in public databases.

The online version of this article includes the following figure supplement(s) for figure 4:

**Figure supplement 1.** Mapping TRLs in single-cell data and databases.

different for cells sorted by 4-1BB positivity versus MART1 recognition. MART1-reactive cells were more likely to be found in exhausted and late-activated clusters than 4-1BB⁺ cells, which tended to map to earlier activation states as well to the NK-like cluster (*Figure 4e*). The latter may be partly due to their prevalence in UM9 (*Figure 4—figure supplement 1c*).

Mapping TCRs from enriched 4-1BB⁺ cells to TILs in corresponding patients from previously published TCR-seq data revealed that a large fraction of TILs in these cultures was among the identified TRL clones (*Figure 4f*). In six of the eight previously published TIL cultures, we were able to

recover and sequence the corresponding cryopreserved biopsy of that patient. Five of the six cultures had many TCRs that overlapped between TILs and biopsies (*Figure 4g* and *Figure 4—figure supplement 1a and b*). UM3 did not have any matching TCRs between the biopsies and expanded TIL cultures. No overlap was observed between samples from different patients with the 4-1BB[+]/MART1[+]-selected TCRs (*Figure 4—figure supplement 1d*). This is reasonable given the different HLA genotypes of these patients. In general, very few clonotypes were shared between samples from different patients (*Figure 4—figure supplement 1e*). In summary, experimentally identified TRLs mapped back to biopsies were overrepresented in proliferative exhausted and *IL7R*[+] exhausted T cells, and these T cells could also be identified in sequenced cultures of expanded TILs.

## Identifying potential antigens recognized by TRLs

To understand potential antigens recognized by a given TCR, we searched for matches in public databases of known TCRβ chains, based on CDR3β motifs, using TCRMatch (*Chronister et al., 2021*). The results confirmed that a number of TCRs matched known MART1-reactive clonotypes but also revealed matching TCRs that might recognize other melanoma antigens, such as PMEL and MAGEA1 (*Figure 4—figure supplement 1f, g, and h*). Overall, a large number of TCRs also matched clonotypes recognizing viral antigens such as influenza, EBV, and human coronaviruses. However, an even greater number lacked known matches, suggesting they might recognize tumor-specific antigens (neoantigens).

However, public databases are currently limited. As an alternative approach to nominate tumor-reactive clonotypes, we used an algorithm designed to predict clusters of TCRs that are sufficiently similar and HLA-restricted to potentially recognize the same antigens. GLIPH2 (*Huang et al., 2020*) nominated a few high-confidence clusters of similar TCR sequences based on the biopsy scRNA-seq data (*Figure 4h* and *Supplementary file 3*). To further test the hypothesis that each cluster recognizes the same antigen, we mapped the CDR3β motifs of these sequences to the same public databases. Interestingly, the highest-ranked cluster was overrepresented for sequences matching MART1, and this cluster also exhibited significant restriction toward HLA-A*02:01, on which MART1 is known to be presented. The remaining high-confidence clusters mainly matched antigens of viral origin or lacked matches to known antigens (*Figure 4h*).

Similar to the experimentally determined MART1-reactive T cells, the bioinformatically determined cluster of potential MART1-recognizing clones mostly belonged to the exhausted CD8[+] T cell cluster (*Figure 4i*, *Figure 4—figure supplement 1i*). This supports the finding that TRLs found in biopsies may have predominantly exhausted profiles. In addition, a high-confidence group of clonotypes co-localized to the NK-like cluster was nominated by GLIPH2, although no sequences from this cluster matched the public TCR-antigen pairs (*Figure 4h*, *Figure 4—figure supplement 1i*). Given that the cells from this cluster were experimentally identified as 4-1BB[+] upon tumor co-culture (*Figure 4b*), one might speculate that they could recognize unknown tumor antigens. Taken together, these findings suggest that both bioinformatically inferred TRLs from biopsies and ex vivo expanded TRLs tend to have exhausted phenotypes. Among these, T cells recognize known melanoma antigens such as MART1, PMEL, and MAGEA1, as well as unknown sequences that might be neoantigens.

## TRLs home to the tumor of PDX mice bearing liver metastases of UM

TIL cultures contain both TRLs, virus-specific T cells, and bystander T cells that grow during IL2-containing culture conditions. We had previously found that TILs can home to subcutaneously growing skin melanoma metastases (*Forsberg et al., 2019*; *Jespersen et al., 2017*) but we were interested in determining whether they could identify and home to UM growing in the liver. The PDX models used to generate sphere cultures were prepared by subcutaneous transplantation. To instead generate a liver metastasis model, we performed splenic injection of UM22 into NOG mice (*Sugase et al., 2020*). Tumor cells forming liver metastases were further transplanted via tail vein injection into hIL2-NOG mice. After verifying tumors by ultrasound, autologous bulk or MART1-selected UM22 TILs were intravenously injected for TIL therapy studies (*Figure 5a*). The mice were harvested 3 weeks later, and tumors recovered from the liver were analyzed by flow cytometry, IHC, scRNA, and TCR-seq. As expected from *Figure 3f and g*, MART1-selected TILs had a higher degree of activation than unselected bulk UM22 TILs (*Figure 5b*), despite similar numbers of TILs present in the livers. There were no visible differences in the size of the liver tumors between mice receiving unselected or

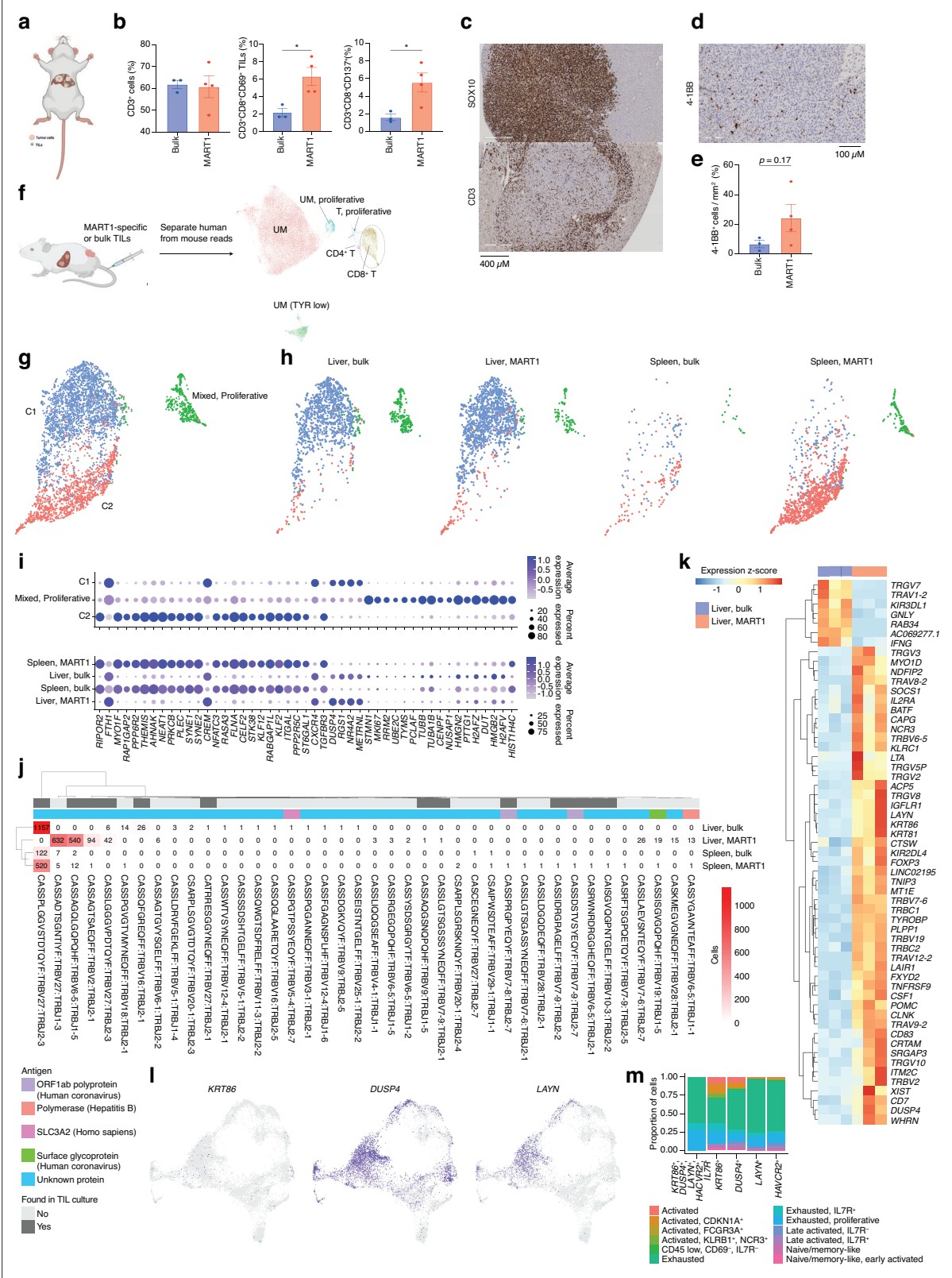

**Figure 5.** CD8 T cells get activated by tumor cells in vivo. (**a**) Bulk unsorted tumor-infiltrating lymphocytes (TILs) or MART1-selected TILs were injected into hIL2-NOG mice carrying liver tumors from patient UM22. (**b**) Flow cytometry analysis of single-cell suspension liver metastasis, comparing treatment of UM22 TILs and UM22 MART1-specific TILs for CD3⁺, CD3⁺CD8⁺CD69⁺, or CD3⁺CD8⁺CD137⁺. (**c**) Immunohistochemical (IHC) with diaminobenzidine (DAB) showing tumor (SOX10) and TILs (CD3) within a liver metastasis, (**d, e**) corresponding analysis of 4-1BB⁺ TILs in a section, (**d**) and in image analysis

*Figure 5 continued on next page*

*Figure 5 continued*

comparing both treatments (**e**). Statistical tests in b and e were unpaired two-tailed t-tests, assuming equal variance. *: p<0.05; **: p<0.01. (**f**) Samples of tumor and TILs from the liver and spleen, respectively, were sequenced with single-cell RNA-seq (scRNA-seq). n=3 biological replicates were performed for each group of liver samples, and n=2 for spleen samples (out of which one spleen sample for MART1-selected TILs was the pooled material of two independent mice). Sequencing reads mapping to human and mouse were separated with XenoCell (*Cheloni et al., 2021*), after which cells were clustered and annotated as described in Methods. (**g**) Subclustering of CD8$^+$ T cells identified three overall clusters, one of which represented a mixed profile of the other two, but with marked cell cycle activity. (**h**) Contributions of the different experimental conditions to each CD8$^+$ T cell cluster. (**i**) Markers distinguishing CD8$^+$ T cell clusters, identified using the FindAllMarkers function of Seurat (the union of the top 25 genes per condition, ranked by log$_2$ fold change). Expression per experimental condition is shown below. (**j**) All TCRβ chains identified in each experimental condition. Subsets found in TIL culture scRNA-seq data are highlighted, as are any matches to antigens in public databases. (**k**) Differentially expressed genes between bulk TIL mixtures or MART1-selected TILs present in the livers of mice. A pseudo-bulk approach was used, summing read counts across all cells within a given replicate, and statistical testing performed with DESeq2 (*Love et al., 2014*). Genes with q<0.05 after Benjamini-Hochberg correction were considered significant. (**l**) Expression of *KRT86*, *DUSP4*, and *LAYN* in biopsy CD8$^+$ T cells. (**m**) Mapping indicated genes to phenotypic clusters from the single-cell RNAseq UMAP in *Figure 2a*. Figure generated with BioRender.com.

The online version of this article includes the following figure supplement(s) for figure 5:

**Figure supplement 1.** Characterization of TRLs in the UM22 PDX model.

**Figure supplement 2.** Marker genes on T cells in the UM22 PDX model.

MART1-selected UM22 TILs. This may be due to T cell exclusion, since most TILs colonized the rim of the tumor (*Figure 5c*). More MART1-specific TILs than unselected TILs were able to get activated, as assessed by the activation markers 4-1BB and CD69 (*Figure 5b, d, and e*). This shows that T cells recognizing a tumor antigen can successfully migrate to liver metastases of UM in mice.

## Determining the phenotypes of TRLs in PDX liver metastases using scRNA- and TCR-seq

Next, we profiled the livers and spleens of mice injected with unselected bulk or MART1-selected UM22 TILs by scRNA- and TCR-seq (*Figure 5f*, *Figure 5—figure supplement 1a*). After bioinformatically separating human- from mouse-derived cells and integrating the data, clearly distinguishable clusters emerged containing UM cells, all characterized by the expression of the differentiation genes *TYR*, *MLANA,* and *PMEL*. Their designation as UM cells was further confirmed by inference of their copy number profiles (*Figure 5—figure supplement 1b*). As expected, all UM cells resided in the liver (*Figure 5—figure supplement 1a*), despite the mice being injected with UM cells via the tail vein. The TILs discovered in these samples constituted three clusters: C1, C2, and a proliferative cluster with a mixed set of cells from the two main clusters (*Figure 5g*). Compared with CD8$^+$ T cell clusters from the biopsies, the clusters found in mice did not separate with respect to the same set of markers. Rather, all clusters expressed markers compatible with late-activated or exhausted cytotoxic cells to some extent (*Figure 5—figure supplement 1c*). These TILs were detectable in both the liver and the spleen, although there was a notable correspondence between, on one hand, liver TILs and C1, and on the other, spleen TILs and C2 (*Figure 5g–i*).

Investigating the TCR clonotypes represented in the four experimental conditions revealed several clones unique to or overrepresented in the liver biopsies of mice with MART1-selected T cells, as compared to unselected TILs or spleen samples (*Figure 5j*). Given that these clones have been sorted to enrich for reactivity toward an antigen known to be presented by UM22 cells, and that they have successfully migrated to the liver tumor, this subset is likely to be strongly enriched for TRLs. Among these, some were also found in previous scRNA-seq data from expanded UM22 TILs (*Figure 5—figure supplement 1d–i*).

Differential expression analysis between this TRL subset and unselected TILs found in the livers of PDX mice revealed statistically significant genes that may be potential markers of the tumor-reactive phenotype (*Figure 5k*, *Figure 5—figure supplement 1j*, *Supplementary file 4*). This included the activation marker *TNFRFS9* (4-1BB) as well as *KIR2DL4*, which has been found to be expressed in certain exhausted subsets of CD8$^+$ T cells (*Figure 2c*; *Li et al., 2019*). Interestingly, this subset also bears a striking resemblance to a recently described group of TIM-3$^+$IL7R$^-$ tumor-reactive tissue-resident memory cells in lung cancer, further characterized by the specific expression of *KRT86*, *DUSP4*, and *LAYN* (*Figure 5—figure supplement 2*; *Clarke et al., 2019*). This set was enriched in patients responding to anti-PD1 therapy (*Clarke et al., 2019*). Cells expressing this marker combination were

also present in biopsy samples, predominantly in exhausted clusters (*Figure 5l and m*), as well as in TIL cultures from UM22 (*Figure 5—figure supplement 2b*). This might suggest avenues for enriching TRLs from biopsies for use with ex vivo expansion and ACT based on their gene expression profiles.

## CD8 cells diminish after tumor eradication in PDX models

Although the UM22 PDX model demonstrated signs of TIL homing and activation, TILs were unable to cure the mice from the tumor, neither when grown subcutaneously (*Figure 3d*) nor in the liver (*Figure 5c*). We therefore developed a liver metastasis model of UM1, which responds to TILs in a subcutaneous PDX model (*Figure 3d*). Injection of TILs into hIL2-NOG mice bearing UM1 liver metastases resulted in a complete clearance or tumors as assessed macroscopically and by ultrasound imaging. To characterize the cellular composition of T cells in the livers and spleens of the PDX mice treated with TILs, we performed scRNA- and TCR-seq. Clustering analysis identified 15 clusters of human cells that were annotated by known marker genes defined by differential expression analysis and supported by literature (*Figure 6b and c*, *Figure 5—figure supplement 1c*, *Supplementary files 5 and 6*). Notably, and at variance with the UM22 model, none of the clusters contained any melanoma cells, confirming that the TILs had eradicated the tumor.

We compared number of cells coming from liver and spleen samples. A total of 20,418 cells were found in liver and 18,208 from spleen (*Figure 6—figure supplement 1a and b*). Strikingly, even though the TILs injected were primarily CD8 positive, the T cells remaining in liver and spleen after tumor eradication were primarily CD4$^+$ positive cells (*Figure 6—figure supplement 1b and c*). Most CD8 cells resided in cluster 11 with fewer cells also in clusters 8 and 12 due to expression of cell cycle genes. Re-clustering of cluster 11 resulted in two subpopulations (11.0 and 11.1) annotated as CD8$^+$ cytotoxic expressing killer cell lectin like receptor G1 (*KLRG1*) and CD8$^+$ cytotoxic expressing higher levels of *CD8A* and *CD8B* (*Figure 6—figure supplement 1d*). These cells also expressed higher levels of *CTLA4, TIGIT,* and *TNFRSF9* (encoding activation marker CD137/4-1BB) and the γδTcell markers *TRGC2/TRGV9*. Cluster 8 expressed high levels of Ki-67 (*MKI67*) assembly factor for spindle microtubules (ASPM), stathmin 1 (*STMN1*), DNA topoisomerase II alpha (*TOP2A*), TPX2 microtubule nucleation factor (*TPX2*) corresponding to the proliferative cellular state. Cluster 12 expressed minichromosome maintenance complex components 2, 6, 7 (*MCM2, MCM6, MCM5, MCM7*), GINS complex subunit 2 (*GINS2*), helicase, lymphoid specific (*HELLS*) that are associated with the proliferation and cell cycle state.

Next, we investigated the CD4$^+$ cells and found that clusters 1, 3, 5, 9 expressed programmed cell death 1 (*PD-1*, also known as *PDCD1*), lymphocyte activation gene 3 (*LAG3*), cytotoxic T lymphocyte-associated protein 4 (*CTLA-4*), T cell immunoreceptor with immunoglobulin and ITIM domains (*TIGIT*), thymocyte selection associated high mobility group box (*TOX*), and eomesodermin (*EOMES*) marker genes that are associated with a dysfunctional or exhausted state and was annotated as a CD4$^+$ exhaustion TOX$^+$, CD4$^+$ exhaustion PTCD1$^+$, CD4$^+$ exhaustion LAG3$^+$, and CD4$^+$ exhaustion TIGIT$^+$, respectively. Three adjacent clusters 0, 2, 10 expressed interleukin 7 receptor (*IL7R*), rhotekin 2 (*RTKN2*), C-C motif chemokine receptor 3 (*CCR3*), and protein tyrosine phosphatase receptor type C (PTPRC, also known as CD45) marker genes corresponding to a memory T cells phenotype and were annotated as a CD4$^+$ memory RTKN2$^+$, CD4$^+$ memory CCR3$^+$, and CD4$^+$ memory PTPRC$^+$, respectively. More liver cells contributed to the CD4$^+$ exhaustion cluster whereas the CD4$^+$ memory cluster was primarily made up of spleen cells (*Figure 6—figure supplement 1b*). Finally, three adjacent clusters 7, 6, 4 were associated with a CD4$^+$ Th17 phenotype that play important role in the inflammatory response expressing the transcription factor retinoic acid orphan receptor (ROR)γt (*RORC*), chemokine CC receptor 6 (*CCR6*), a killer cell lectin-like receptor B1 (*KLRB1*, also known as *CD161*), interleukin 4-induced 1 (*IL4I1*), cyclin-dependent kinase inhibitor 1A (*CDKN1A*), aquaporin 3 (*AQP3*) genes (*Figure 6—figure supplement 1e*) and annotated as a CD4$^+$ Th17 CD161$^+$ (KLRB1), CD4$^+$ Th17 PTPN13$^+$, and CD4$^+$ Th17 PTPRM$^+$ respectively.

Finally, we investigated the TCR clonotype abundance and visualized top 10 the most abundant clonotypes (*Figure 7a*). Clonotypes 302, 11155, and 21797 were the most abundant, and as expected, they were present in CD4$^+$ cells. Clonotypes 302 mapped to the CD4$^+$ exhaustion site, 111555 to the CD4$^+$ memory, and 21797 to the CD4$^+$ Th17 cluster (*Figure 7b*). To investigate if there were any differences in gene expression between these clonally expanded T cells in liver versus spleen, we performed pseudo-bulk analysis. Genes such as *FASLG, ICOS, TIGIT, IL2RA* that were higher in T cells

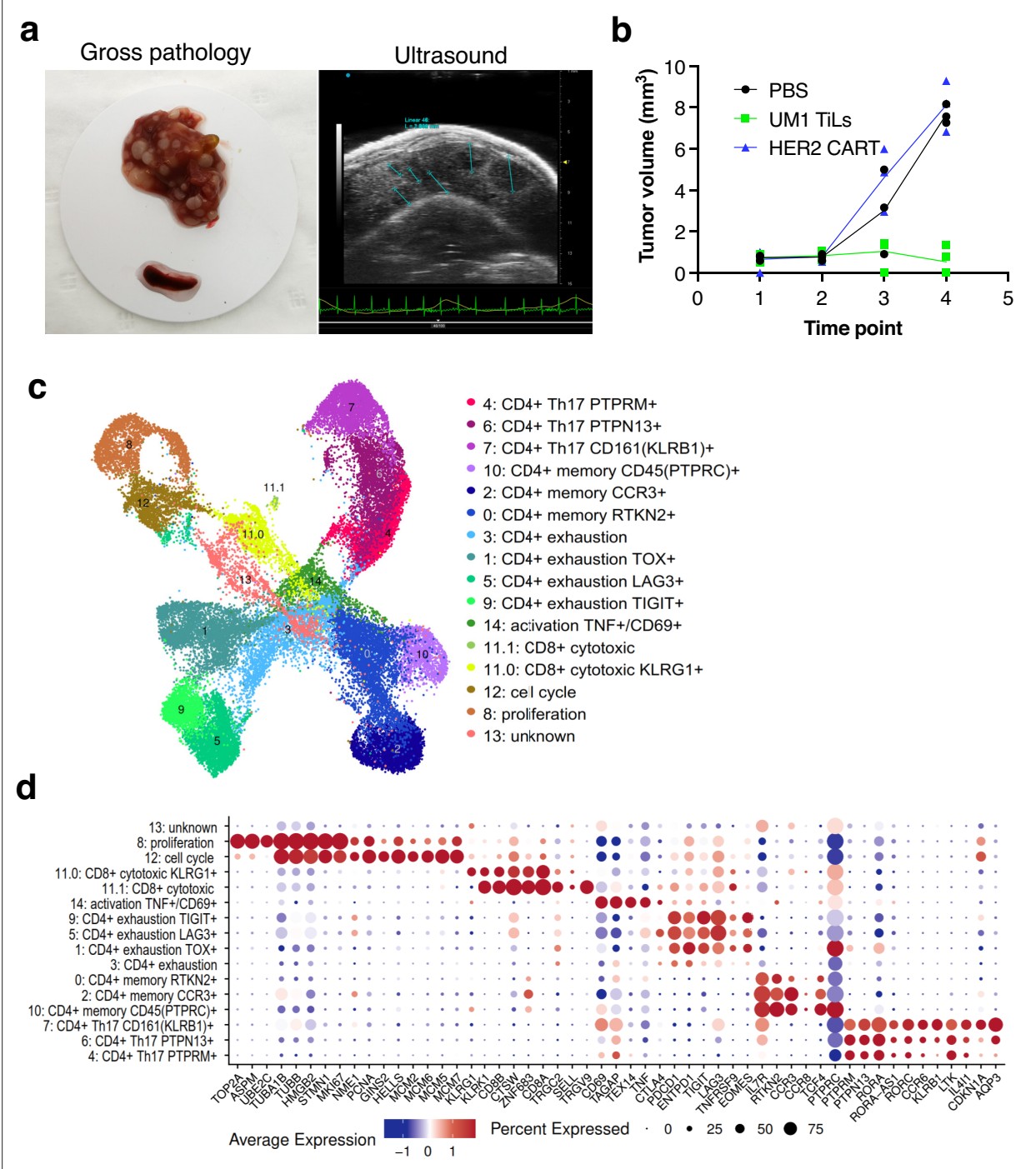

**Figure 6.** Characterization of CD8 T cells in the UM1 PDX model following tumor eradication. (**a**) Establishment of another uveal melanoma (UM) liver metastasis model. UM1 patient-derived xenograft (PDX) cells were injected in the tail vein after having been serially transplanted in spleen followed by harvesting from liver. Ultrasound confirmed growth in liver before injection of autologous UM1 tumor-infiltrating lymphocytes (TILs) or HER2 CAR-T cells as controls. (**b**) Response to TILs as assessed by ultrasound monitoring. (**c**) Uniform Manifold Approximation and Projection (UMAP) of T cells in the UM1 liver metastatic model, showing 15 different cell populations. A list of marker genes was used to annotate clusters. The marker genes list was compiled from differential expression analysis and literature. (**d**) Dot plot showing an average expression of marker genes and detection rate of cells in which the marker gene is detected across 15 cell populations.

The online version of this article includes the following figure supplement(s) for figure 6:

**Figure supplement 1.** Characterization of TILs in vivo in the UM1 PDX.

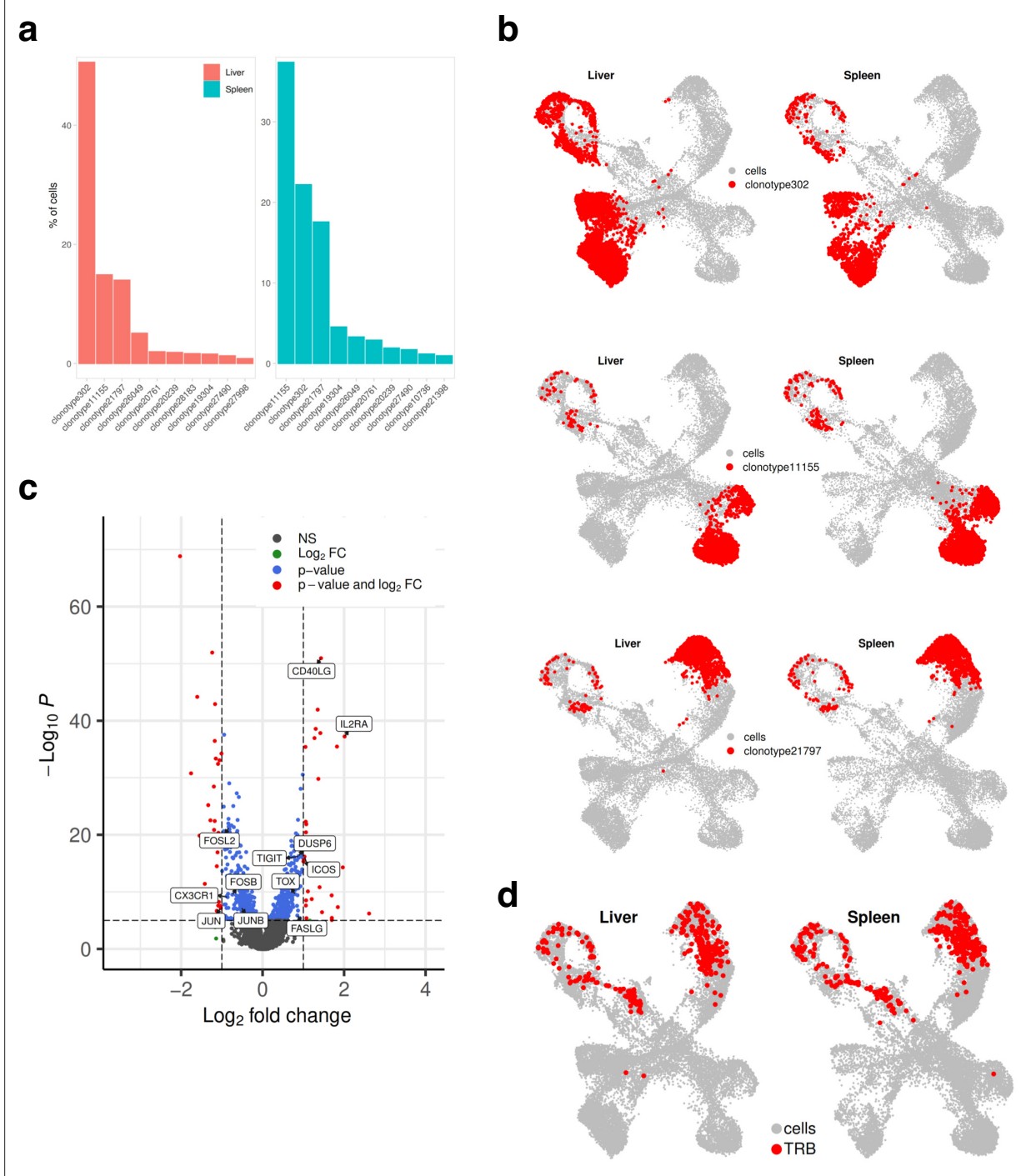

**Figure 7.** Mapping clonotypes in single-cell data from the UM1 PDX model. (**a**) Bar plot showing top 10 the most abundant clonotypes for liver and spleen samples detected by the TCR-seq. (**b**) Uniform Manifold Approximation and Projection (UMAP) showing where clonotypes 302, 11155, and 21797 are residing. (**c**) Volcano plot highlighting differentially expressed genes between liver and spleen detected for the clonotype 302 using pseudo-bulk approach. log₂FC positive means liver is upregulated relative to the control (spleen). (**d**) UMAP showing where tumor-reactive T lymphocytes (TRLs) (from *Figure 3f and g*, *Figure 4*) are residing.

residing in liver are known to be expressed during activation and exhaustion (*Figure 7c*). We also mapped where TRLs are residing by extracting the top 10 T cell receptor beta sequences obtained from the CD137-sorted UM1 TILs (*Figure 3f–g* and *Figure 4*). We identified a few of these in the PDX data and mapped them on the UMAP (*Figure 7d*). TRLs were residing on the CD4[+] Th17 and proliferation/cell cycle clusters.

## Discussion

UM continues to be a condition with unmet medical need, despite recent advances in locoregional therapy and immunotherapy (*Fu et al., 2022*). However, the fact that a T cell engager is prolonging survival in patients with UM having an HLA-A2 genotype (*Nathan et al., 2021*) and that immune checkpoint inhibitors in various combination therapies on occasion can yield clinical benefit (*Piulats et al., 2021*; *Pelster et al., 2021*; *Ny et al., 2021*) suggests that if we learn more about the tumor immunology of UM, then we can devise new immunotherapies. Here, we generated datasets and models that can be used not only to characterize tumor immunity, but also to functionally test hypotheses and targets in autologous culture and in vivo systems.

ACT with TILs has also been tested in patients with metastatic UM (*Chandran et al., 2017*). Although there were responses in a few patients, the data clearly show room for improvement. The HAITILS trial (NCT04812470) addresses whether hepatic arterial infusion results in local accumulation of TILs in liver metastases. However, this will not circumvent the low mutational burden of UM (other than iris melanoma; *Karlsson et al., 2020*) and the potentially low number of TRLs in the cell product (that we see in this study). A greater understanding of TRLs in patients' metastases is therefore of interest to learn how to enhance the activity of TIL products and how to expand, for instance, neoantigen- or melanoma-associated antigen-specific TRLs (*van den Berg et al., 2020*; *Seiter et al., 2002*). Here, we aimed to identify tumor-reactive lymphocytes in terms of marker characteristics and functional assessments of TILs. These analyses suggest that some of the TILs in our cultures that became activated (4-1BB positive) when binding tumor cells were already TRLs (4-1BB or PD-1 positive) when they resided in the tumor. However, they were never able to clear the tumor, and most likely became exhausted or dysfunctional in the microenvironment.

Presence of TRL-specific markers adds validity to a strategy whereby enrichment of TRLs could be done directly from the tumor by sorting for those markers. Indeed, previous studies have employed activation, exhaustion, or tissue-resident markers such as 4-1BB (*Ye et al., 2014*), PD-1 (*Inozume et al., 2010*; *Gros et al., 2014*; *Zheng et al., 2022*), CD39 (*Zheng et al., 2022*; *Simoni et al., 2018*), and CD103 (*Webb et al., 2010*; *Duhen et al., 2018*), or the specificity of a TCR for a melanoma-associated or melanoma-specific antigen (*Yee et al., 2002*; *Dudley et al., 2002*). However, a risk with this strategy is that some TRLs are missed. Most notably, we revealed TRLs in clusters of T cells negative for activation and exhaustion markers. These cells are interesting since they express high levels of perforin and granzyme B, suggesting that they are cytotoxic. It is tempting to speculate that these cells are effector or effector memory cytotoxic T cells since they express RNA encoding CD16, FCRL6, and CX3CR1 (*Clémenceau et al., 2008*; *Schreeder et al., 2008*; *Gerlach et al., 2016*). In the UM9 sample, most of the T cells in the biopsy that mapped to TRLs identified in the tumor sphere assay resided in this cluster.

PDX models have been developed for UM, but to our knowledge, this is the first time they have been used to study immunotherapy using autologous TILs and single-cell sequencing. Although one model responded to TIL therapy, the other model did not. In the UM22 liver metastasis model, we found evidence that T cells were pushed out of the tumor. This may partly explain the failure of immune checkpoint inhibitors in many patients (*Algazi et al., 2016*) and why ACT with TILs does not work for the majority of patients with metastatic UM (*Chandran et al., 2017*). Nevertheless, we profiled TRLs that preferentially migrated to liver metastases in mice, as opposed to non-reactive T cells and those that migrate to the spleen. We identified marker genes and phenotypic states that might be used to identify relevant subsets present in patient biopsies and TIL cultures. These T cells appeared to resemble a recently described *IL7R[-]HAVCR2[+]KRT86[+]DUSP4[+]LAYN[+]* subset, which has been associated with response to anti-PD1 therapy in lung cancer (*Clarke et al., 2019*). They also tended to express *ITGAE* (CD103), suggesting that they may be tissue-resident memory cells. This subset was identified in both PDX liver metastases, cultured TILs, and patient biopsies. The phenotypes of cells with this marker combination were predominantly exhaustion-like in the PDX samples as

well as in biopsies. This is compatible with the results from the profiling of biopsy CD8+ T cells, where mapping of experimentally identified MART1-reactive T cells back to biopsies and computational prediction of tumor antigen-recognizing T cells suggested an overrepresentation of TRL candidates in exhausted and late-activated gene expression-based clusters. Whether cells expressing these canonical exhaustion-related markers are truly dysfunctional or capable of eliciting an antitumor response remains to be explored (*Clarke et al., 2019*).

In the UM1 liver metastasis model, the tumor was eradicated, demonstrating that TILs can cause a complete response also of liver metastases. The single-cell data obtained confirmed the loss of melanoma cells but it also resulted in a loss of CD8+ T cells. It is well known in immunology that the fate of CD8 T cells after an acute immunological reaction is either death of the effector cells or development of memory. A strength of the PDX experiment is that we could model. But since we harvested the livers after tumor eradication, a limitation is that we could not investigate the expression of genes in T cells that are actively combating the tumor. However, since the CD4+ T cells were present in high numbers, we could compute differences between the T cells of the same clonotype residing either in liver or in spleen. This analysis did indeed suggest that some level of activation/exhaustion remained even after tumor clearance. Moreover, some of the TCRs from the co-culture experiment of tumor-spheres and TILs designed to reveal TRLs were indeed identified in vivo among the CD8 and the proliferative cells. We are therefore confident that PDX models such as UM22 and UM1 will be useful in future studies where T cells are sorted for potential markers prior to expansion or infusion, where the T cells are genetically engineered or grown in cocktails of factors to enhance potency and stemness. We also propose the model to be appropriate for further humanization to unravel the interplay between tumor cells, liver cells and stroma cells, myeloid cells, and other immune cells.

When mapping TCRs to TCR databases, we noted not only similarities to known melanoma-associated antigens, but also to viruses. Importantly, TCRs that recognize different antigens can share CDR3β chains. The TCR alpha chain is even more promiscuous but not always reported in these databases, which makes it difficult to find accurate TCR pair matches. Therefore, until current databases have been expanded and improved, we cannot be certain about the exact antigen matches of some of these receptors. Nevertheless, an increasing body of literature suggests that bystander cells in tumors are often antigen-experienced, can be directed against bacteria or viruses, and even participate in tumor control, for instance during immune checkpoint inhibitor therapy (*Rosato et al., 2019*; *Çuburu et al., 2022*; *Hu et al., 2022*). Whether the presence of foreign antigen-specific T cells is beneficial for cellular immunotherapy remains to be evaluated.

In summary, we profiled the tumor microenvironment of metastatic UM and identified the phenotypes of tumor-reactive T cell subsets. The results of this study provide new avenues for targeted expansion of T cells to improve cellular and immune checkpoint immunotherapy. Importantly, however, to potentiate TIL therapies in UM, immune cold tumors likely need to be turned hot, i.e., to create an inflammatory environment promoting T cell infiltration. Simultaneously, lymphodepletion is needed to avoid existing T cells sequestering IL2 needed for the TIL infusion product. Tumor-targeting treatments such as IHP or percutaneous hepatic perfusion may fulfill these needs and could act as adjuvant therapy for both immune checkpoint inhibitor treatment and TIL therapy. As melphalan triggers immunogenic death, influx of new T cells into cold tumors may occur, partly explaining the very high response rate of these therapies (*Olofsson Bagge et al., 2023*). Ongoing clinical trials are combining locoregional therapy with ICI (NCT04463368) to capitalize on this concept. However, data from these trials have not yet been published. Combination of liver-directed therapy and TIL therapy is also warranted in future animal experiments and clinical trials. We propose that PDX models are optimal to capture the intra-patient heterogeneity of cancer and that multiple mechanisms in tumor immunology and treatment can be modeled at single-cell resolution.

## Methods

### Patient samples

According to the ethical approval of the Regional Ethical Review Board (#289-12, #44-18, and #144-13), the patients provided oral and written information with signed informed consent agreements. Biopsies were extracted from either subcutaneous or liver metastases (*Supplementary file 7*). Tumor biopsies were divided into pieces that were snap-frozen, minced, embedded in formalin blocks, or

used for TIL cultures and cryopreservation. Snap-frozen tumor fragments were homogenized using a Bullet Blender (Next Advance, NY, USA). DNA and RNA were extracted using the AllPrep DNA/RNA kit (QIAGEN, Germany).

## Immunostainings of sections

FFPE patient samples were evaluated using H&E staining and IHC. IHC was performed using an auto-stainer (Autostainer Link 48, Dako) with primary anti-human antibodies against Melan-A (clone A103; Dako), S100 (IR504; Dako), PRAME (E7I1B, CST), SOX10 (E6B6I, CST), HMB45 (M0634; Dako), CD3 (D4V8L, CST), and CD137 (E6Z7F, CST). HRP DAB or HRP Magenta (Dako) was used to stain the proteins of interest, and counterstaining was performed using hematoxylin.

We also performed multiplex immunofluorescence using The PhenoCycler-Fusion system (Akoya Biosciences) at the Spatial Proteomics Facility at SciLifeLab, Stockholm. The data for CD3, ICOS, and PD-1 primary antibodies are shown here; however, the full dataset will be published elsewhere.

## PDX models

Animal experiments were conducted in conformity with EU directive 2010/63 (Regional Animal Ethics Committee of Gothenburg approvals no. 2014-36 and 2018-1183). PDX models were generated by transplanting small pieces into the flank (subcutaneous) of 6- to 8-week-old female immunocompromised, non-obese, severe combined immune deficient interleukin-2 chain receptor γ knockout mice (NOG mice) or hIL2-NOG mice (Taconic, Denmark). Cryopreserved single cells from biopsies were thawed and transplanted by splenic injection to create liver metastases in NOG mice or hIL2-NOG mice (Taconic, Denmark). All PDX tumors were analyzed using IHC, and their identity was verified by staining patient biopsies using clinically graded antibodies. For subcutaneous implants, tumor growth was monitored twice a week following TILS treatment. After establishment, the liver metastasis models formed liver metastases equally well by splenic injection and tail vein injection. The mice were evaluated using ultrasound scans before TIL treatment and grossly assessed at the time of sacrifice.

## Generation of TILs and MART1-specific T cells

The metastatic fragments were placed in culture medium (90% RPMI 1640 [Invitrogen], 10% heat inactivated human AB serum [HS, Sigma-Aldrich], 6000 IU/ml recombinant human IL2 [Peprotech], penicillin and streptomycin [Invitrogen]) to generate young TILs (yTILs) as previously described (*Jespersen et al., 2017*). MART1-specific yTILs were identified as previously described (*Karlsson et al., 2020*) and sorted using FACSAria III (BD Biosciences). yTILs were expanded using a standard REP (Rapid Expansion Protocol), constituting irradiated (40 Gy) allogeneic feeder cells, CD3 antibody (clone: OKT3) (30 ng/ml) (Miltenyi), medium (AIM-V, Invitrogen) supplemented with 10% HS and 6000 IU/ml IL2 and monitored for 14 days with media changes (*Jespersen et al., 2017*). After completion of the REP, MART1 specificity was confirmed using MART1-specific dextramers (Immudex, Copenhagen, Denmark). REP and REP MART1 TILs were used for downstream, cell line, spheroid, and in vivo experiments.

## Spheroid assay

PDX models were harvested, and tumor tissue was digested using a human tumor dissociation kit (Miltenyi) according to the manufacturer's instructions in conjunction with the gentleMACS Octo tissue dissociator (Miltenyi). This single-cell suspension was used to prepare spheroids using an ultra-low attachment, 96-well, clear, round-bottomed plates (Corning, NY, USA), and GBM-MG serum-free media (CLS Cell Lines Service, Germany) for 72–96 hr in a 37°C incubator. To monitor treatment responses, cells were labeled with CellVue Jade Cell Labeling Kit (Thermo Fisher Scientific) and/or CellVue Claret Far Red Fluorescent Cell Linker Mini Kit (Sigma) and used for co-culture experiments. The 3D images were captured using IncuCyte (Essen BioScience)±TILs treatment, followed by imaging and analysis at 72 hr with IncuCyte Live Cell Analysis. Tumor spheres stained with a far-red dye were measured for the red object area/μm (*Buder et al., 2013*) mask with >100 μm. Flow cytometry was used to confirm the positively stained immune cells (Jade-FITC channel) and cancer cells (far-red APC channel).

## Co-culture and TCR-seq

A human UM cell line derived from patient UM22 (*Karlsson et al., 2020*) was grown in complete medium (RPMI-1640 supplemented with 10% FBS, glutamine, and gentamicin) and cultured at

37°C with 5% $CO_2$. The cell line or spheroids were analyzed with or without TILs, and the co-culture was analyzed using human CD3 (FITC and HIT3a, BioLegend), CD8 (APC and SK1, BioLegend), CD107a (APC and H4A3, BD Biosciences), CD137 (PE and 4B4-1, BioLegend), active caspase-3 (clone C92-605; BD Biosciences), and granzyme B (clone GB11; BD Biosciences) antibodies for 30 min at 4°C. Flow cytometry data were acquired using the BD Accuri C6 and BD Accuri C6 Plus (BD Biosciences). The supernatants of these samples were subjected to Granzyme B secretion analysis using an ELISA kit (R&D Systems). Following detection of the activated cell populations in certain samples, a microfluidics-based flow cell sorting instrument (WOLF NanoCellect, San Diego, CA, USA) was used to sort $CD3^+CD137^+$ cells from the co-culture of tumor spheres and TILs. The RNA of the sorted cells was extracted using a Nucleospin RNA XS kit (Macherey-Nagel), and its concentration was measured using QuBit. Libraries were prepared for TCR profiling using a SMARTer Human TCR a/b Profiling Kit v2 (catalog number 63478) and subsequent NGS library preparation and bead purification; Tapestation 4200 (Agilent) was used to confirm sample quality control with automated electrophoresis.

## Statistical analysis for spheroid assays

Statistical tests for the spheroid assays were done using two-tailed unpaired t-tests, with p-values represented as * for $p<0.05$ and ** for $p<0.01$; all error bars represent standard error (SEM), unless otherwise stated.

## scRNA-seq data analysis

### Library preparation

5' GEX and TCR-seq: Single-cell suspensions prepared from cryopreserved UM biopsies were subjected to dead-cell removal using beads (Miltenyi Biotech) before quality and viability validation and loading on Chromium instrument (10x Genomics). Single-cell 5' gene expression and TCR libraries were prepared using vendor protocol with the Next GEM Single Cell 5' Kit v2 (10x Genomics, catalog number 1000263), Library Construction Kit (catalog number 1000190), Single Cell Human TCR Amplification Kit (catalog number 1000252), and Dual Index kit TT Set A (catalog number 1000215). Libraries were sequenced for quality check on iSeq100 (Illumina) before deep sequencing on NovaSeq 6000 (Illumina) in the required format to yield at least 20,000 reads per cell for gene expression and 5000 reads per cell for TCR libraries.

### Alignment and gene expression quantification

Paired scRNA-seq and TCR-seq data from biopsies were aligned to the 10x Genomics-provided GRCh38 reference genome (v. 2020A) and V(D)J reference (v. 7.0.0) using the Cell Ranger pipeline (v. 7.0.1) and the 'multi' command, with both 'Gene Expression' and 'VDJ-T' analyses activated, as well as with the setting 'check-library-compatibility,false'.

### Removal of ambient RNA

To remove ambient RNA, CellBender (v. 0.2.0) (*Fleming et al., 2022*) was used on the output from Cell Ranger, with the parameters '`--fpr 0.01 --epocs 150`' together with expected and total cell numbers estimated from Cell Ranger quality control plots, as described in the instructions at https://cellbender.readthedocs.io/en/latest/usage/index.html (accessed September 15, 2022). The values for the parameters '`--low-count-threshold`' and '`--learning-rate`' were changed from their defaults for some problematic samples and iteratively adapted as needed after inspecting CellBender output quality control plots, as described in the documentation.

### Separation of human and mouse reads in PDX scRNA-seq data

To separate reads of mouse and human origin in the scRNA-seq data from PDX samples, XenoCell (v. 1.0) (*Cheloni et al., 2021*; *Conway et al., 2012*) was used. As input, compressed fastq files, merged across lanes, were provided, together with reference genomes for human and mouse (the same genomes as for Cell Ranger). An index was created with the command 'xenocell.py generate_index', with the parameters '`--threads 1 --memory 24`'. Reads were classified according to host (mouse) or graft (human) with the command 'xenocell.py classify_reads' and the parameters '`--barcode_start 1 --barcode_length 16 --threads 1 -memory 24 --compression_level 1`'. Reads belonging

to human and mouse, respectively, were then extracted into new fastq files using the command 'xeno-cell.py extract_cellular_barcodes' with the same values for barcode_start, barcode_length, threads, memory, and compression_level parameters as in the previous step, but with '--lower_threshold 0 --upper_threshold 0.1' for graft and '--lower_threshold 0.9 --upper_threshold 1.0' for host. The workflow was implemented in a Nextflow script available on GitHub, (copy archived at *Karlsson, 2022*). The new fastq files were then used for separate Cell Ranger runs for human and mouse reads for a given sample, as described above for human biopsy samples, except with the corresponding 10x Genomics-provided mouse reference genome instead for reads determined to be of mouse origin.

## Quality control and cell filtering

Gene counts and assembled TCR sequences were imported to R with Seurat (*Satija et al., 2015*). An initial object was created and normalized with the NormalizeData function of the same package. Samples were integrated with fastMNN (*Haghverdi et al., 2018*) (batchelor package v. 1.10.0) via the convenience function RunFastMNN from the SeuratWrappers (v. 0.3.0) package (parameters: 'features = 4000, auto.merge=T, d = 150, k = 15'). Assembled and annotated TCR sequences from Cell Ranger were further added to this object using djvdj (*Sheridan, 2020*). Cells with <200 reads were then removed.

A first set of potential doublets were identified using scDblFinder (v. 1.8.0), using default parameters. In addition to this, cells expressing conflicting lineage-associated markers were also marked as doublets. Incompatible marker sets were defined as any among CD3D, CD4, CD8A, CD8B or with assembled TCR receptor expressed together with any of MLANA, PMEL, TYR, HBA1, HBA2, or HBB; any cell with assembled TCR receptor expressed together with any of CD14, CD19, MS4A1, or JCHAIN; or any among MLANA, PMEL or TYR expressed together with CD14, CD19, MS4A1, JCHAIN, or NCR1. For the purpose of further refining doublet detection, an initial clustering was performed using the Seurat functions FindNeighbors (parameters: 'reduction = 'mnn', dims = 1:150, k.param=5') and FindClusters ('resolution = 5'). For each cluster, a two-sided Fisher's exact test was carried out, assessing whether that cluster was associated with doublets. p-Values were adjusted for multiple testing with Benjamini-Hochberg correction. In addition, each cluster was also assessed for higher than expected UMI counts per cell. Doublet clusters to be removed were nominated from any cluster with Fisher's test odds ratio <1 and q<0.05, percent doublets greater than or equal to the 95% percentile of doublet percentages, or UMI counts above the 95% percentile. These thresholds were chosen in conjunction with visual inspection of plots regarding how these candidate clusters associated with other metrics of poor quality, such as proximity in UMAPs to cells expressing incompatible lineage markers and higher/lower than usual ribosomal or mitochondrial read percentage.

Global thresholds to keep cells fulfilling the following criteria were set: number of detected genes >200, UMI counts >500, ribosomal read percentage <40%, and percentage of genes in the top 100 expressed genes <80%. However, further specific filtering was made for two cell types, erythrocytes and plasma cells, known to have generally lower RNA content. Since common global thresholds on RNA and number of detected genes may filter out these cell types entirely, lower thresholds were motivated specifically for these. To establish suitable thresholds, candidate members of these cell types were first identified based on the distinctly expressed marker genes HLA1 (erythrocytes) and JCHAIN (plasma cells, with IL3RA and LILRA4 additionally demanded to be zero), respectively. Cells in each respective group (without having removed candidate doublets) were clustered based on Spearman correlation coefficients with the pheatmap function in the R package of the same name (default parameters) and visualized together. In both cases, cells were divided into clear clusters, some of which had predicted doublets were overrepresented. The latter clusters also tended to have higher than expected RNA content. Cluster without overrepresented candidate doublets were retained and the rest discarded from the dataset. Thresholds to keep cells with UMI counts and number of detected genes greater than the first quartile were defined for each of the two cell types based on cells in clusters not associating with doublets.

Further to this, additional quality filtering thresholds for remaining cells were determined adaptively using miQC (v. 1.2.0) (*Hippen et al., 2021*), which first creates mixture models to assess the distribution of number of genes detected versus percentage of mitochondrial reads in each sample and then filters cells based on a posterior probability of being compromised derived from this distribution. For

a few samples that failed to converge with the default model, a one-dimensional mixture model was used instead. For all samples, the parameters 'posterior_cutoff = 0.9' and 'keep_all_below_boundary = TRUE' were used to filter out low-quality cells.

## Sample integration, unsupervised clustering, and cell-type annotation

After removing low-quality cells and doublets, all samples were reintegrated using fastMNN, with the same parameters as described above. A UMAP embedding was created using the Seurat function RunUMAP ('parameters: reduction = 'mnn', dims = 1:150'). Unsupervised clustering was performed with FindNeighbors ('reduction = 'mnn', dims = 1:150') and FindClusters ('resolution = 2'). Genes differing between clusters were assessed using the FindAllMarkers function (default parameters). Clusters were annotated based on a combination of generally known cell-type markers (CD3D and CD3G for T cells, CD8A for CD8$^+$ T cells, CD4 for CD4$^+$ T cells, FOXP3 for regulatory T cells, NCAM1 and NCR1 for NK cells, CD19 and MS4A1 for B cells, HBA1, HBA2, and HBB for erythrocytes, CD14 for general monocytic cells, PMEL, MLANA, and TYR for melanocytic cells, as well as MKI67 for actively cycling cells) and markers compiled from literature (*Li et al., 2019*; *Durante et al., 2020*; *Malone, 2021*; *Collin and Bigley, 2018*; *Buonomo et al., 2022*). CD8$^+$ T cells were determined and studied in greater detail to identify further subgroups. After subsetting the main Seurat object to only contain CD8$^+$ T cells, these were subclustered by rerunning FindNeighbors ('reduction = 'mnn', k.param = 15, dims = 1:150') and FindClusters ('algorithm = 1, resolution = 1.3'). A new UMAP was created using RunUMAP ('reduction = 'mnn', dims = 1:150, min.dist = 0.05, n.neighbors = 15'). 16 clusters were obtained, among which some were merged based on similar expression of key marker genes. The new CD8$^+$ T cell clusters were annotated based on a combination of literature-derived markers (*Li et al., 2019*; *Zhang et al., 2021*; *Zhao et al., 2020*; *Crespo et al., 2013*) for CD8$^+$ T cell phenotypes as well as genes found to distinguish the clusters that could be related to specific phenotypes.

## Processing of scRNA-seq from TIL cultures

scRNA-seq data of TIL cultures from a previous study (*Karlsson, 2022*) were obtained, loaded into a Seurat object in R, and normalized with the NormalizeData function (default parameters). Variable features were determined using FindVariableFeatures ('selection.method = 'vst', nfeatures = 4000'). Cell cycle scores were then calculated using CellCycleScoring (default provided gene sets) and data scaled using ScaleData ('vars.to.regress=c('S.Score', 'G2M.Score')'). Principal components were calculated using the RunPCA function (npcs = 30) and a UMAP embedding calculated using the RunUMAP function (dims = 1:30). These steps were performed on each sample separately, without sample integration.

## GLIPH2 analysis

TCR sequences were clustered into inferred specificity groups using GLIPH2 (*Huang et al., 2020*). As input, unique TCR sequences and their counts from annotated CD8$^+$ T cells were given, together with previously published HLA genotypes from bulk RNA and WGS (*Karlsson, 2022*). HLA genotypes for two additional samples were inferred from previously published exome data (*Ny et al., 2021*) using nf-core/hlatyping (v2.0.0) (*Ewels, 2022*), with the parameter '--genome GRCh37'. Clonotypes with more than one TCR alpha chain were excluded from analysis, as these are not supported by GLIPH2. Parameters provided were the option to use the v2 human reference for CD8 T cells and with 'all_aa_interchangeable' set to Yes. The tool was run from the web interface and according to instructions at http://50.255.35.37:8080/ (accessed December 21, 2022). High-confidence TCR clusters were considered those with V-gene bias score <0.05, at least three unique TCRs and at least three unique individuals contributing TCRs.

## Copy number analysis

To determine copy number changes in UM cells from scRNA-seq data, the inferCNV R package (v. 1.10.1) (*Tickle et al., 2019*) was used according to the instructions provided at https://github.com/broadinstitute/infercnv/wiki (accessed November 2, 2022). A genomic position file was created from the reference annotation provided with Cell Ranger (v. 2020A) using the inferCNV helper script gtf_to_position_file.py. All cells annotated as UM were marked as 'malignant' and remaining cells were

used as a reference diploid background as input to inferCNV, together with the following parameters: 'cutoff = 0.1, cluster_by_groups = TRUE, denoise = TRUE, HMM = FALSE, num_threads = 1, analysis_mode='samples', output_format = 'pdf'.

## Differential expression analysis

Differentially expressed genes between experimental conditions in scRNA-seq data from PDX samples were assessed using a pseudo-bulk approach together. First, all the human CD8[+] T cells for each sample were used to create gene subset expression matrix. All cells in a given sample were summed on a per-gene basis to create pseudo-bulk samples, imitating a bulk RNA-seq experiment. These samples were normalized and analyzed for differential expression between the conditions represented by the samples using DESeq2 (*Love et al., 2014*), with the parameters test = 'LRT', reduced = '~1', relative to a full model specification encoding the two experimental conditions to be compared. For each pairwise comparison of conditions, only samples in those two conditions were included in the DESeq2 main object. To extract a table of results, the 'results' function from the same R package was used, with the parameter 'alpha = 0.05'. Significant genes were visualized with the pheatmap function from the R package of the same name (v. 1.0.12), with the parameter 'scale = 'row'.

## scRNA- and TCR-seq analysis for UM1 sample

FASTQs files lane-1 and lane-2 were concatenated and used as an input for the XenoCell (v. 1.0) tool to split scRNA-seq data on host (mm) and graft (hs). XenoCell classify_reads function was used to calculate the percentage of graft- and host-specific reads for each cellular barcode. Then, extract_cellular_barcodes function was used to extract cellular barcodes for the host (mm), and the same function was used to extract cellular barcodes for the graft (hs). The feature-barcode matrix for graft (hs) was generated using Cell Ranger multi (v. 7.0.1) pipeline. Parameters `--localcores` and `--localmem` were set to 32 and 240 respectively and human reference genome GRCh38 (v. 2020-A) and V(D)J reference (v. 7.0.0) were used. Next, CellBender remove-background CellBender (v. 0.2.0) function was used to remove ambient RNA. For each sample CellBender parameters were adjusted as follows `--expected-cells` (25L: 7573; 25S: 2274; 39L: 7973; 39S: 10527; 45L: 8438; 45S: 10447; 55L: 6318; 55S: 4143), `--total-droplets-included` (25L: 17000; 25S: 10000; 39L: 12000; 39S: 17000; 45L: 16000; 45S: 19000; 55L: 15000; 55S: 13000), `--fpr` (25L: 0.01; 25S: 0.01; 39L: 0.01; 39S: 0.01; 45L: 0.01; 45S: 0.01; 55L: 0.01; 55S: 0.01), `--epochs` (25L: 150; 25S: 150; 39L: 200; 39S: 150; 45L: 150; 45S: 200; 55L: 150; 55S: 150), `--low-count-threshold` (25L: 3; 25S: 5; 39L: 3; 39S: 3; 45L: 3; 45S: 5; 55L: 5; 55S: 5), `--learning-rate` (25L: 0.0000125; 25S: 0.00005; 39L: 0.00000625; 39S: 0.0000125; 45L: 0.00000625; 45S: 0.00000625; 55L: 0.00005; 55S: 0.00005).

R (v. 4.3.2) was used for downstream analysis. Data generated by CellBender was parsed using Read_CellBender_h5_Multi_Directory function from scCustomize (v. 2.0.1) package. Seurat object was created using CreateSeuratObject function from Seurat (v. 5.0.1) package with parameters min. cells and min.features set to 3 and 200 respectively. This resulted in the Seurat object with 23,884 genes across 64,091 cells. V(D)J data was added to the Seurat object using import_vdj function from djvdj (v. 0.1.0) library. Doublets were removed using scDblFinder function from scDblFinder (v. 1.16.0) package that resulted in 57,200 cells. Cells with a low coverage were removed before using scDblFinder (nCount<200, *Germain, 2024*). QCs filtering was performed to remove dying/dead, any remaining empty droplets or multiples cells. Cells with a high gene deletion number could indicate doublets or multiplets, thus, cells that have unique gene counts over 6500 or less than 400 were removed. Cells with a total read count (or UMI) less than 500 and more than 25,000 were filtered out. Finally, cells with mitochondrial counts more than 20% and ribosomal counts more than 15% were removed. Next, the data was normalized, scaled, and dimensionality reduction applied using NormalizeData, FindVariableFeatures (nfeatures = 2000), ScaleData, RunPCA. This step followed standardized Seurat workflow ('Data analysis workflow, Integrative analysis in Seurat v5', https://satijalab.org/seurat/articles/seurat5_integration, accessed 2024, *Satija, 2024*). Integration was performed using IntegrateLayers function from Seurat (v. 5.0.1) package with parameters set to method = CCAIntegration, orig.reduction = 'pca', new.reduction = 'integrated.cca'. FindNeighbors (reduction = 'integrated. cca', dims = 1:70) and FindClusters (resolution = c(0.1–2.0), cluster.name = 'cca_clusters') functions from Seurat (v. 5.0.1) package were used and resolution 0.6 (15 clusters) was selected. FindAllMarkers (logfc.threshold = 0.3, min.pct = 0.3, only.pos = TRUE, test.use = 'wilcox') function from Seurat (v.

5.0.1) package was used to find differentially expressed genes. To visualize results DimPlot, VlnPlot, FeaturePlot, and DotPlot Seurat (v. 5.0.1) functions were used. For plotting the volcano plot we used EnhancedVolcano (*Blighe, 2018*); for plotting a bar plot ggplot2 (v. 3.5.0) was used. To pseudo-bulk, AggregateExpression function from Seurat (v. 5.0.1) package was used. The gene list was compiled using marker genes coming from differential expression analysis and from the literature. Results were visualized as a heatmap using plotGroupedHeatmap function from the scater (v. 1.30.1) package. To find clonotype abundance calc_frequency function from djvdj (v. 0.1.0) package was used; to visualize results plot_clone_frequency function was used from the same package.

## TCR sequence analysis

### Bulk TCR-seq data analysis

TCR sequences sequenced with the Takara SMARTer Human TCR a/b Profiling Kit v2 were aligned and assembled using the open source Cogent NGS Immune Profiler software (v. 1.0) provided by the same company, using the parameters '-r TCR' and '-t Both'.

### Matching of TCR receptors between samples

TCRs from scRNA-seq samples and/or bulk TCR-seq samples were matched based on identical combinations of CDR3β, Vb and Jb. This was the minimal common combination of criteria available on which to consistently compare TCRs across all TCR-sequenced samples, since the bulk TCR-seq protocol does not provide information in pairing of alpha and beta chains.

### Matching of TCR receptors to public databases

To match TCRs to public databases with information on possible antigens they might recognize, TCRMatch (v. 1.0.2) (*Chronister et al., 2021*) was used (parameter: -s 0.97) together with a modified version of the reference database from IEDB (*Vita et al., 2019*) provided with the tool, which we expanded with additional information from VDJdb (*Shugay et al., 2018*), McPAS-TCR (*Tickotsky et al., 2017*), and TCR3d (*Gowthaman and Pierce, 2019*) (using the datasets for both cancer and virus-targeting TCRs). In addition, this database file was further modified to replace internal commas in antigen names, since the default formatting conflicts with the use of comma as the main delimiter in the database file. This facilitated easier analysis of the results. TCR matches were visualized with Sankey diagrams using the ggsankey R package (*Sjoberg, 2021*).

### Overrepresentation analysis for TCRs in CD8+ T cell clusters

To assess whether the 41BB- or MART1 reactivity experiment would enrich for clones mapping to any phenotype relative to cultured TILs in general, a binomial test was used. The test considers the observed fraction of cells with matches to the reactivity screen versus an expected probability. Given that cells from the reactivity experiment were taken from a TIL culture, we wanted to assess whether these had a meaningfully different distribution compared to cultured TILs in general in regard to cluster representation. Therefore, the expected probability was estimated as the fraction of TILs that match any given cluster. Thus, these two frequencies were compared with one two-sided binomial test (binom.test in R) per cluster, and p-values further adjusted for multiple testing with Bonferroni correction.

## Acknowledgements

We thank Carina Karlsson for providing technical support. Grant support came from Cancerfonden (to JAN and ROB), Familjen Erling Persson (to JAN), Knut and Alice Wallenberg Foundation (to JAN and ROB), Vetenskapsrådet (to JAN and ROB), Sjöbergstiftelsen (to JAN), BioCARE Strategic grants (to JAN), Lion's Cancerfond Väst (to SA and LN), Västra Götaland Regionen ALF grant (to JAN, ROB, and LN), Assar Gabrielsson fond (to VS), Gustaf V Jubileumsklinikens forskningsfond (to LN) and Wilhelm & Martina Lundgrens Vetenskapsfond (to JK). JAN is the Inaugural Chair of Melanoma Discovery, primarily supported by donations to Perkins from family and friends of Scott Kirkbride, the MACA Cancer 200 ride for cancer research, the HBF Foundation and Perpetual. The Genomics WA facility (AS) is supported by BioPlatforms Australia, State Government Western Australia, Australian Cancer Research Foundation, Cancer Research Trust, Harry Perkins Institute of Medical Research, Telethon

Kids Institute and the University of Western Australia. We also thank the Pawsey Supercomputing Centre for providing computational resources for this project.

## Additional information

### Funding

| Funder | Grant reference number | Author |
|---|---|---|
| Cancerfonden | | Roger Olofsson Bagge<br>Lars Ny<br>Jonas A Nilsson |
| Erling-Persson Family Foundation | | Jonas A Nilsson |
| Vetenskapsrådet | | Roger Olofsson Bagge<br>Jonas A Nilsson |
| Sjöbergstiftelsen | | Jonas A Nilsson |
| Harry Perkins Institute of Medical Research | start-up grants | Jonas A Nilsson |
| Västragötaland Region | ALF grant | Jonas A Nilsson<br>Roger Olofsson Bagge<br>Lars Ny |
| Knut and Alice Wallenberg Foundation | | Jonas A Nilsson<br>Roger Olofsson Bagge |
| BioCARE | strategic grants | Jonas A Nilsson |
| Lion's Cancerfond Väst | | Samuel Alsen<br>Lars Ny |
| Assar Gabrielssons Foundation | | Vasu R Sah |
| Gustaf V Jubileumsklinikens forskningsfond | | Lars Ny |
| Wilhelm och Martina Lundgrens Vetenskapsfond | | Joakim W Karlsson |

The funders had no role in study design, data collection and interpretation, or the decision to submit the work for publication.

### Author contributions

Joakim W Karlsson, Data curation, Software, Formal analysis, Validation, Investigation, Visualization, Writing – original draft, Writing – review and editing, Conducted all bioinformatic analysis except for the UM1 liver metastasis model data, generated figures and wrote several sections of the paper; Vasu R Sah, Formal analysis, Validation, Investigation, Methodology, Writing – review and editing, Conducted functional experiments, generated figures and wrote sections of the paper; Roger Olofsson Bagge, Resources, Writing – review and editing, Coordinated patient consents, surgeries and collection of liver metastases of uveal melanoma; Irina Kuznetsova, Data curation, Formal analysis, Validation, Investigation, Writing – review and editing, Conducted bioinformatic analysis for the liver metastatic model of the UM1 sample; Munir Iqba, Methodology, Writing – review and editing, Conducted single-cell sequencing; Samuel Alsen, Methodology, Writing – review and editing, Performed cell sorting and flow analyses; Sofia Stenqvist, Investigation, Methodology, Writing – review and editing, Generated PDX models and developed new transplantation and imaging techniques; Alka Saxena, Resources, Supervision, Writing – review and editing, Managed single-cell sequencing workflow; Lars Ny, Resources, Writing – review and editing, Performed patient recruitment and supervision; Lisa M Nilsson, Conceptualization, Resources, Supervision, Investigation, Methodology, Project administration, Writing – review and editing, Supervised, managed the biobank and sequencing, generated TILs,

managed, conceived and visualized data from animal experiments; Jonas A Nilsson, Conceptualization, Data curation, Supervision, Funding acquisition, Investigation, Visualization, Writing – original draft, Project administration, Writing – review and editing, Conceived and supervised the study, wrote the paper and generated figures

### Author ORCIDs
Roger Olofsson Bagge ⓘ https://orcid.org/0000-0001-5795-0355
Alka Saxena ⓘ https://orcid.org/0000-0001-5683-0618
Jonas A Nilsson ⓘ https://orcid.org/0000-0003-0346-6837

### Ethics
Human subjects: According to the ethical approval of the Regional Ethical Review Board in Gothenburg (#289-12, #44-18 and #144-13), the patients provided oral and written information with signed informed consent agreements.
Animal experiments were conducted in conformity with E.U. directive 2010/63 (Regional Animal Ethics Committee of Gothenburg approvals no. 2014-36 and 2018-1183).

Reviewer #1 (Public Review): https://doi.org/10.7554/eLife.91705.3.sa1
Reviewer #2 (Public Review): https://doi.org/10.7554/eLife.91705.3.sa2
Author response https://doi.org/10.7554/eLife.91705.3.sa3

## Additional files

### Supplementary files
• Supplementary file 1. Statistically identified marker genes for each CD8+ T cell cluster in *Figure 2a*, using the FindMarkers function in the Seurat R package.

• Supplementary file 2. Complete statistics for the analysis of cluster representation among reactivity-screened TCRs matched to biopsy CD8+ T cells, as compared to representation among tumor-infiltrating lymphocytes (TILs) matched to the same clusters (*Figure 4d*).

• Supplementary file 3. Complete statistical output from the GLIPH2 analysis referred to in *Figure 4h and i* and *Figure 4—figure supplement 1i*.

• Supplementary file 4. Complete statistics for the differential expression analyses in *Figure 5k* and *Figure 5—figure supplement 1j*.

• Supplementary file 5. Differentially expressed genes in different clusters of the Uniform Manifold Approximation and Projection (UMAP) in *Figure 5*.

• Supplementary file 6. References for marker gene selection and annotation of clusters in *Figure 5*.

• Supplementary file 7. Description of the patient samples analyzed in this study.

• MDAR checklist

### Data availability
All newly generated single-cell RNA/TCR sequencing data are available at NCBI Sequence Read Archive under the collective accession number PRJNA1135711. Single-cell RNA/TCR sequencing data from TILs were from a previous study and are available via controlled access from European Genome-Phenome Archive (Dataset ID EGAD00001006031).

The following dataset was generated:

| Author(s) | Year | Dataset title | Dataset URL | Database and Identifier |
|---|---|---|---|---|
| Karlsson J, Sah VR, Bagge RO, Kuznetsova I, Iqbal M, Alsén S, Stenqvis S, Saxena A, Ny L, Nilsson LM | 2024 | Patient-derived xenografts and single-cell sequencing identifies three subtypes of tumor-reactive lymphocytes in uveal melanoma metastases | https://www.ncbi.nlm.nih.gov/bioproject/PRJNA1135711 | NCBI BioProject, PRJNA1135711 |

The following previously published dataset was used:

| Author(s) | Year | Dataset title | Dataset URL | Database and Identifier |
|---|---|---|---|---|
| Joakim K, Jonas AN | 2020 | Sequencing of uveal melanoma metastases | https://ega-archive.org/datasets/EGAD00001006031 | European Genome-phenome Archive, EGAD00001006031 |

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
