## [Editor Report · eLife assessment]

This study presents **valuable** findings on tumor-reactive T cells in liver metastases of uveal melanoma (UM). The authors conducted single-cell RNA sequencing to identify potential tumor-reactive T cells and used PDX models for functional analysis. The evidence supporting their claims is **solid**. The work will be of interest to scientists working in the field of uveal melanoma.

---

## [Referee Report · Reviewer #1 (Public Review)]

This work successfully identified and validated TRLs in hepatic metastatic uveal melanoma, providing new horizons for enhanced immunotherapy. Uveal melanoma is a highly metastatic cancer that, unlike cutaneous melanoma, has a limited effect on immune checkpoint responses, and thus there is a lack of formal clinical treatment for metastatic UM. In this manuscript, the authors described the immune microenvironmental profile of hepatic metastatic uveal melanoma by sc-RNAseq, TCR-seq, and PDX models. Firstly, they identified and defined the phenotypes of tumor-reactive T lymphocytes (TRLs). Moreover, they validated the activity of TILs by in vivo PDX modeling as well as in vitro co-culture of 3D tumorsphere cultures and autologous TILs. Additionally, the authors found that TRLs are mainly derived from depleted and late-activated T cells, which recognize melanoma antigens and tumor-specific antigens. Most importantly, they identified TRLs-associated phenotypes, which provide new avenues for targeting expanded T cells to improve cellular and immune checkpoint immunotherapy.

Comments on revised manuscript

The revised manuscript has addressed all my concerns.

---

## [Referee Report · Reviewer #2 (Public Review)]

Summary:

The study's goal is to characterize and validate tumor-reactive T cells in liver metastases of uveal melanoma (UM), which could contribute to enhancing immunotherapy for these patients. The authors used single-cell RNA and TCR sequencing to find potential tumor-reactive T cells and then used patient-derived xenograft (PDX) models and tumor sphere cultures for functional analysis. They discovered that tumor-reactive T cells exist in activated/exhausted T cell subsets and in cytotoxic effector cells. Functional experiments with isolated TILs show that they are capable of killing UM cells in vivo and ex vivo.

Strengths:

The study highlights the potential of using single-cell sequencing and functional analysis to identify T cells that can be useful for cell therapy and marker selection in UM treatment. This is important and novel as conventional immune checkpoint therapies are not highly effective in treating UM. Additionally, the study's strength lies in its validation of findings through functional assays, which underscores the clinical relevance of the research.

Weaknesses:

The manuscript may pose challenges for individuals with limited knowledge of single-cell analysis and immunology markers, making it less accessible to a broader audience.

---

## [Author Response]

The following is the authors’ response to the original reviews.

**Reviewer #1 (Public Review):**
Summary:This work successfully identified and validated TRLs in hepatic metastatic uveal melanoma, providing new horizons for enhanced immunotherapy. Uveal melanoma is a highly metastatic cancer that, unlike cutaneous melanoma, has a limited effect on immune checkpoint responses, and thus there is a lack of formal clinical treatment for metastatic UM. In this manuscript, the authors described the immune microenvironmental profile of hepatic metastatic uveal melanoma by sc-RNAseq, TCR-seq, and PDX models. Firstly, they identified and defined the phenotypes of tumor-reactive T lymphocytes (TRLs). Moreover, they validated the activity of TILs by in vivo PDX modelling as well as in vitro coculture of 3D tumorsphere cultures and autologous TILs. Additionally, the authors found that TRLs are mainly derived from depleted and late activated T cells, which recognize melanoma antigens and tumor-specific antigens. Most importantly, they identified TRLs associated phenotypes, which provide new avenues for targeting expanded T cells to improve cellular and immune checkpoint immunotherapy.Strengths:Jonas A. Nilsson, et al. has been working on new therapies for melanoma. The team has also previously performed the most comprehensive genome-wide analysis of uveal melanoma available, presenting the latest insights into metastatic disease. In this work, the authors performed paired sc-RNAseq and TCR-seq on 14 patients with metastatic UM, which is the largest single-cell map of metastatic UM available. This provides huge data support for other studies of metastatic UM.

We thank the reviewer for these kind words about our work.

Weaknesses:Although the paper does have strengths in principle, the weaknesses of the paper are that these strengths are not directly demonstrated. That is, insufficient analyses are performed to fully support the key claims in the manuscript by the data presented. In particular:The author's description of the overall results of the article should be logical, not just a description of the observed phenomena. For example, the presentation related to the results of TRLs lacked logic. In addition, the title of the article emphasizes the three subtypes of hepatic metastatic UM TRLs, but these three subtypes are not specifically discussed in the results as well as the discussion section. The title of the article is not a very comprehensive generalization and should be carefully considered by the authors.

We thank the reviewer for the critical reading of our work. We have added more data and more discussion.

The authors' claim that they are the first to use autologous TILs and sc-RNAseq to study immunotherapy needs to be supported by the corresponding literature to be more convincing. This can help the reader to understand the innovation and importance of the methodology.

We have gone through the manuscript and found that we only refer to being first in using PDX models and autologous TILs to study immunotherapy responses by single-cell sequencing. While there are data to be deduced from other studies, we still believe this to be an accurate statement.

In addition, the authors argue that TILs from metastatic UM can kill tumor cells. This is the key and bridging point to the main conclusion of the article. Therefore, the credibility of this conclusion should be considered. Metastatic UM1 and UM9 remain responsive to autologous tumors under in vitro conditions with their autologous TILs.

UM1 responds also in vivo in the subcutaneous model in the paper. We have also finished an experiment where we show that this model also responds in a liver metastasis model. These data have been added in this revised version of the paper. We add two main figures and one supplementary figure where we characterize the response in vivo and also by single-cell sequencing of TILs.

In contrast, UM22, also as a metastatic UM, did not respond to TIL treatment. In particular, the presence of MART1-responsive TILs. The reliability of the results obtained by the authors in the model of only one case of UM22 liver metastasis should be considered. The authors should likewise consider whether such a specific cellular taxon might also exist in other patients with metastatic UM, producing an immune response to tumor cells. The results would be more comprehensive if supported by relevant data.

The reviewer has interpreted the results absolutely right, the allogenic and autologous MART1-specific TILs cells while reactive in vitro against UM22, cannot kill this tumor either in a subcutaneous or liver metastases model. We hypothesize this has to do with an immune exclusion phenotype and show weak immunohistochemistry that suggest this. We hope the addition of more UM1 data can be viewed as supportive of tumor-reactivity also in vivo.

In addition, the authors in that study used previously frozen biopsy samples for TCR-seq, which may be associated with low-quality sequencing data, high risk of outcome indicators, and unfriendly access to immune cell information. The existence of these problems and the reliability of the results should be considered. If special processing of TCR-seq data from frozen samples was performed, this should also be accounted for.

We agree with the reviewers and acknowledge we never anticipated the development of single-cell sequencing techniques when we started biobank 2013. We performed dead cell removal before the 10x Genomics experiment. We have also done extensive quality controls and believe that the data from the biopsies should be viewed as a whole and that quantitative intra-patient comparisons cannot be done.

**Reviewer #2 (Public Review):**
Summary:The study's goal is to characterize and validate tumor-reactive T cells in liver metastases of uveal melanoma (UM), which could contribute to enhancing immunotherapy for these patients. The authors used single-cell RNA and TCR sequencing to find potential tumor-reactive T cells and then used patientderived xenograft (PDX) models and tumor sphere cultures for functional analysis. They discovered that tumor-reactive T cells exist in activated/exhausted T cell subsets and in cytotoxic effector cells. Functional experiments with isolated TILs show that they are capable of killing UM cells in vivo and ex vivo.Strengths:The study highlights the potential of using single-cell sequencing and functional analysis to identify T cells that can be useful for cell therapy and marker selection in UM treatment. This is important and novel as conventional immune checkpoint therapies are not highly effective in treating UM. Additionally, the study's strength lies in its validation of findings through functional assays, which underscores the clinical relevance of the research.

We thank the reviewer for these kind words about our work.

Weaknesses:The manuscript may pose challenges for individuals with limited knowledge of single-cell analysis and immunology markers, making it less accessible to a broader audience.

The first draft of the manuscript (excluding methods) was written by a person (J.A.N) who is not a bioinformatician. It has been corrected to include the correct nomenclature where applicable but overall it is written with the aim to be understandable. We have made an additional effort in this version.

**Reviewer #1 (Recommendations For The Authors):**
(1) Firstly, the authors should provide high-resolution pictures to ensure readability for readers.

We have converted to pdf ourselves and that improved resolution. We are happy to provide high-resolution to the office if needed for the printing.

(2) Furthermore, some parts of the article are more colloquial, and the authors should consider the logic and academic nature of the overall writing of the article. For example, authors should double-check whether the relevant expressions in the results are correct. For example, 'TCR' in the fourth part of the results should be 'TRLs'.

We thank the reviewer for the recommendations and have gone through the manuscript.

(3) Moreover, UM22 is described several times in the results as a metastatic UM and should be clearly defined in the methodology.

The UM22 and UM1 samples are described in-depth in Karlsson et al., Nature Communications, 2020, a paper that is cited in the beginning of Results as part of the narrative. The current work can be viewed as an extension of that work.

(4) Finally, it is recommended that authors describe a part of the results in full before citing the corresponding picture, otherwise, it will lead to confusion among readers.

We have made an effort in the revised version to describe the new data in more detail.

**Reviewer #2 (Recommendations For The Authors):**
The manuscript is very interesting and important to understanding key aspects of uveal melanoma immune profile and functionality. However, in my opinion, there are a few aspects that could be addressed.- The manuscript lacks comprehensive details about the samples used, such as their disease progression, response to treatment, or any relevant information that could shed light on potential differences between samples. It would be valuable to know whether these samples were collected before any systemic treatment or if any of the patients underwent immunotherapy post-sample collection, along with the outcomes of such treatments. Providing this information would enrich the manuscript and provide a more holistic view of the research.

We thank the reviewer for the recommendation and have included a new Supplementary table 7 with information about the samples. We have also pasted in individual samples’ contribution to the UMAP to add further holistic view.

- The results presented and discussed in the manuscript seem to indicate that there were no significant differences across the various samples, including comparisons between lymph-node and liver metastases. However, this lack of variation or the reasons for not discussing any observed differences should be clarified. If there are distinctions between the samples, it would be beneficial to discuss these findings in the manuscript.

We thank the reviewer for the recommendation. Whereas 14 samples are many for a uveal melanoma study it is not really powered to do intra-patient comparisons.

- The manuscript may pose difficulties for individuals with limited knowledge of single-cell analysis and immunology markers, potentially limiting its accessibility. To make the research more inclusive, the authors might consider presenting the technical aspects of their work in a less descriptive manner and providing explanations for those less familiar with the technology. This would help a broader audience grasp the significance of the study's findings.

The manuscript is from a multidisciplinary team where all have read and commented. The draft was written by a tumor biologist and edited by a bioinformatician for accuracy. We honestly think it is more understandable than most studies in this bioinformatics era. But we have tried to describe the new data in an easier way.